# DENSITY ESTIMATION WITH LLMS: A GEOMETRIC INVESTIGATION OF IN-CONTEXT LEARNING TRAJECTORIES

**Toni J.B. Liu**
Department of Physics
Cornell University, USA
`jl3499@cornell.edu`

**Raphaël Sarfati**
School of Civil and Environmental Engineering
Cornell University, USA
`raphael.sarfati@cornell.edu`

**Nicolas Boullé**
Department of Mathematics
Imperial College London, UK
`n.boulle@imperial.ac.uk`

**Christopher J. Earls**
Center for Applied Mathematics
School of Civil and Environmental Engineering
Cornell University, USA
`earls@cornell.edu`

## ABSTRACT

Large language models (LLMs) demonstrate remarkable emergent abilities to perform in-context learning across various tasks, including time-series forecasting. This work investigates LLMs' ability to estimate probability density functions (PDFs) from data observed in-context; such density estimation (DE) is a fundamental task underlying many probabilistic modeling problems. We use Intensive Principal Component Analysis (InPCA) to visualize and analyze the in-context learning dynamics of LLaMA-2, Gemma, and Mistral. Our main finding is that these LLMs all follow similar learning trajectories in a low-dimensional InPCA space, which are distinct from those of traditional density estimation methods such as histograms and Gaussian kernel density estimation (KDE). We interpret these LLMs' in-context DE process as a KDE algorithm with adaptive kernel width and shape. This custom kernel model captures a significant portion of the LLMs' behavior despite having only two parameters. We further speculate on why the LLMs' kernel width and shape differ from classical algorithms, providing insights into the mechanism of in-context probabilistic reasoning in LLMs. Our codebase, along with a 3D visualization of an LLM's in-context learning trajectory, is publicly available at `https://github.com/AntonioLiu97/LLMICL_inPCA`.

## 1 INTRODUCTION

Modern Large Language Models (LLMs) showcase surprising emergent abilities that they were not explicitly trained for (Brown et al., 2020; Dong et al., 2024), such as learning from demonstrations (Si et al., 2023) and analogies in natural language (Hu et al., 2023). Such capacity to extract patterns directly from input text strings, without relying on additional training data, is generally referred to as in-context learning (ICL).

Recently, LLMs have been shown to achieve competitive performance in various mathematical problems, including time series forecasting (Gruver et al., 2024), inferring physical rules from dynamical systems (Kantamneni et al., 2024; Liu et al., 2024), and learning random languages (Bigelow et al., 2024). To solve these tasks, an LLM must possess some capacity for probabilistic modeling — the ability to infer conditional or unconditional probability distribution structures by collecting relevant statistics from in-context examples (Akyürek et al., 2024).

We investigate LLMs' ability to perform density estimation (DE), which involves estimating the probability density function (PDF) from data observed in-context. Our core experiment is remarkably straightforward. As illustrated in Figure 1, we prompt LLMs such as LLaMA-2 (Touvron et al., 2023), Gemma (GemmaTeam et al., 2024), and Mistral (Jiang et al., 2023) with a series of data

points $\{X_i\}_{i=1}^n$ sampled independently and identically from an underlying distribution $p(x)$. We then observe that the LLMs' predicted PDF, $\hat{p}_n(x)$, for the next data point gradually converges to the ground truth as the context length $n$ (the number of in-context data points) increases.[1]

To interpret the internal mechanisms (Olsson et al., 2022; Bietti et al., 2023; Dai et al., 2023; von Oswald et al., 2023) underlying an LLM's in-context DE process, we use Intensive Principal Component Analysis (InPCA) (Teoh et al., 2020; Quinn et al., 2019; Mao et al., 2024) to embed the estimated PDF at each context length $\{\hat{P}_n\}$ in the probability space (Figure 2). Our visualizations reveal that the in-context DE processes of the tested LLMs all follow similar low-dimensional paths, which are distinct from those of traditional density estimation methods like histograms and Gaussian kernel density estimation (KDE).

By studying the geometric features of these in-context DE trajectories, we identify a strong bias towards Gaussianity, which we argue is a telltale feature of kernel-based density estimation (Rosenblat, 1956; Wand & Jones, 1994; Silverman, 2018). This observation inspires us to model an LLM's in-context DE process as a kernel density estimator with adaptive kernel. Despite having only two parameters, kernel shape and width, this bespoke KDE model captures the in-context learning trajectories with high precision.

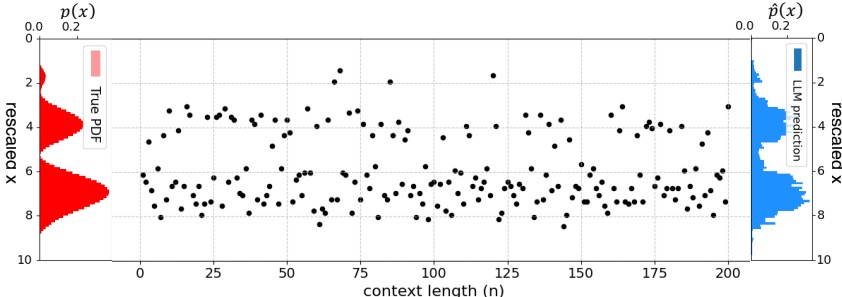

Figure 1: In-context density estimation experiment. LLaMA-2 13b is prompted with 200 numbers sampled from $p(x)$ (left in red) (Appendix A.9), and predicts the PDF $\hat{p}(x)$ (right in blue) for the next number.

**Main contributions.**

1. We introduce a framework for probing an LLM's ability to estimate unconditioned PDFs from in-context data.

2. We apply InPCA to analyze the low-dimensional features in the in-context learning trajectories of various LLMs.

3. We interpret the LLMs' in-context DE algorithm as a form of adaptive kernel density estimation, providing evidence for a dispersive induction head mechanism (Olsson et al., 2022; Akyürek et al., 2024).

## 2 BACKGROUND

**In-context learning of stochastic dynamical systems.** Gruver et al. (2024) pioneered LLM-based time-series prediction by prompting an LLM with time-series data serialized as comma-delimited, multi-digit numbers. They observed competitive predictions for future states encoded as softmax probabilities. Their data serialization scheme proved simple yet effective, allowing for easy log-likelihood evaluation of predicted next numbers. It has since become a common prompting method (Jin et al., 2024; Requeima et al., 2024; Zhang et al., 2024; Liu et al., 2024) for evaluating LLMs' numerical abilities.

Building on this prompting technique, Liu et al. (2024) tested LLMs' ability to in-context learn the transition rules of various stochastic and chaotic dynamical systems. They introduced a recursive algorithm, termed Hierarchy-PDF, to extract an LLM's predicted PDF for the next data point based

---

[1]Our prompts consist purely of comma-delimited multi-digit numbers, without any natural language instructions.

on histories observed in-context: $\hat{p}(x_t|X_0, \ldots, X_{t-1})$. This allowed them to rigorously compare the predicted PDF against the ground-truth transition probability $p(x_t|x_{t-1})$. Our work leverages the data serialization technique proposed by Gruver et al. (2024) and extracts LLMs' probabilistic predictions using the Hierarchy-PDF algorithm introduced by Liu et al. (2024).

**Density Estimation.** DE (Izenman, 1991) is a long-standing statistical problem with both classical (Silverman, 2018; Rosenblat, 1956; Scott, 1979; Lugosi & Nobel, 1996; Parzen, 1962) and modern machine-learned solutions (Papamakarios et al., 2021; Sohl-Dickstein et al., 2015). It underlies the learning of more complex stochastic processes. For example, in the first-order Markov process studied in (Liu et al., 2024; Bigelow et al., 2024; Zekri et al., 2024), the transition probability $p(x_t|x_{t-1})$ has to be estimated in context. This can be viewed as a conditional density estimation problem, where for each possible value of $x_{t-1}$, the density of $x_t$ must be estimated. In other words, learning a Markov process involves performing multiple density estimations; one for each conditioning state. Our current focus on unconditional density estimation thus serves as a stepping stone towards understanding these more complex learning tasks.

## 3 METHODOLOGY

Here, we explain how we prompt an LLM to perform density estimation over in-context data, and how we extract and analyze its in-context learning trajectories with InPCA. Our methodology consists of 5 steps, the first 3 of which are illustrated in Fig. 2.

**1. Sampling and prompt generation.** Starting from a ground-truth probability density function $p(x)$, we generate a series of independent samples from it $X_0, \ldots, X_{t-1}$. Consistent with Gruver et al. (2024), we then serialize and tokenize this series into a text string consisting of a sequence of comma-delimited, 2-digit numbers, forming a prompt that looks like "6 1 , 4 2 , 5 9 , ... , 8 1 , 3 2 , 5 8 , " (Figure 2 (a)).

**2. Extracting estimated densities with *Hierarchy-PDF*.** Upon prompting with such a text string, we read out the LLM's softmax prediction over the next token, yielding probabilities for 10 tokens (0-9)[2], creating a coarse, 10-bin PDF spanning $x \in (0, 100)$. We then read out the next token, which refines one of the tens bins by further dividing it into 10 ones bins. This process is repeated recursively 10 times, until all bins are refined, yielding a predicted PDF for the next state $p(x_t)$, which is a discrete PDF object consisting of $10^N$ bins, where $N$ is the number of digits used in representing each number.[3] We interpret the extracted PDF $\hat{p}(x_t)$ as the LLM's estimation of the ground-truth $p(x)$.

**3. Visualizing DE trajectories with InPCA.** When there are very few in-context data, an LLM's estimated $\hat{p}_0(x)$ is close to a uniform distribution over the domain $(0, 100)$, showing a state of neutral ignorance, which is a reasonable Bayesian prior (Xie et al., 2022). However, as the number of in-context data ($n$) increases, the LLM gathers more information about the underlying distribution. As a result, the estimated $\hat{p}_n(x)$ gradually converges to the ground truth $p(x)$ (Figure 2 (b)). The series of estimated PDFs $\hat{p}_0(x), \hat{p}_1(x), ..., \hat{p}_n(x)$ over context length $n$ forms what we term the "in-context DE trajectory". These trajectories, we argue, offer important clues about how the LLM performs in-context learning.

However, with our 2-digit representation, each $\hat{p}(x)$ is a vector living in a 99-dimensional space[4], making direct analysis of the trajectory challenging. We therefore use InPCA to embed these PDF objects into a low-dimensional Euclidean space (Figure 2 (c)), and then analyze their geometric features.[5]

**4. Comparing with classical DE algorithms.** To interpret the low-dimensional geometric features revealed by InPCA, we embed the DE trajectories of well-known algorithms — specifically, kernel density estimator (KDE) and Bayesian histograms — in the same two-dimensional (2D) space. Surprisingly, we find that the DE trajectories of LLMs, KDE, and Bayesian histograms are simultane-

---

[2]The logits of all other tokens are ignored.

[3]We use the Hierarchy-PDF algorithm (Liu et al., 2024), which performs this recursive search efficiently for transformers.

[4]$10^2 - 1$, where the $-1$ dimension is from the normalization constraint that $\int p(x)dx = 1$.

[5]The idea of using InPCA to visualize the training dynamics of machine learning systems has been explored in Quinn et al. (2019); Mao et al. (2024), which is further discussed in Appendix A.2

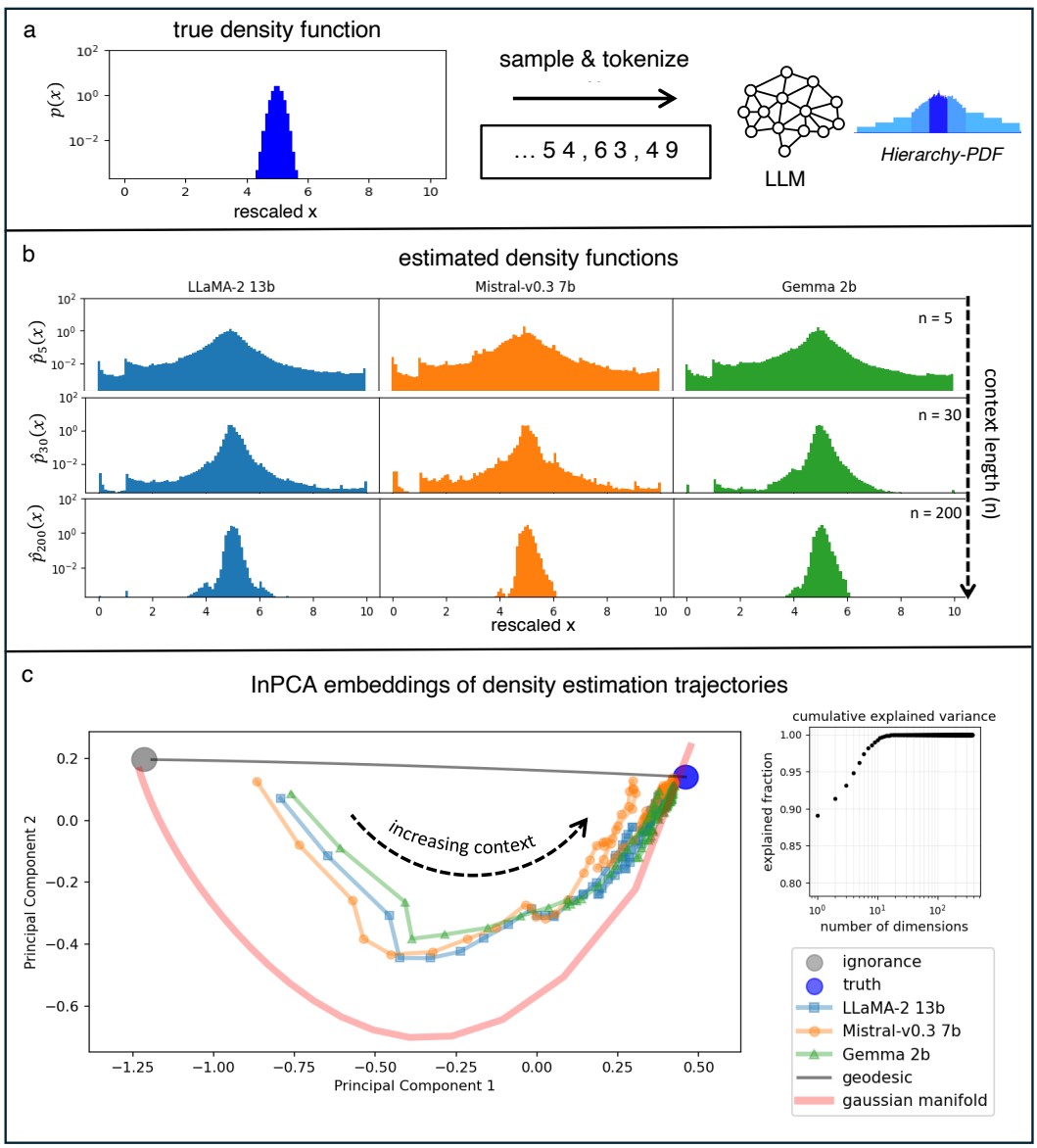

Figure 2: **Visualization pipeline of LLMs' in-context density estimation process.** (a) Data points are independently sampled from a ground truth distribution (Gaussian in this example), then serialized as comma-delimited, two-digit numbers to prompt LLMs, such as LLaMA-2, Mistral-v0.3, and Gemma. (b) *Hierarchy-PDF* extracts each LLM's estimated density function $\hat{p}_n$ at each context length $n$. (c) InPCA reveals low-dimensional structures in density estimation trajectories, capturing 91% of pairwise Hellinger distances in 2 dimensions. Visual guides: gray - uniform PDF representing maximal ignorance, deep blue - ground truth PDF, and pink - 1D submanifold of centered Gaussians with variances ranging from $\infty$ to 0. All LLMs investigated in this work exhibit similar DE trajectories, which are geometrically bounded between the geodesic and the Gaussian sub-manifold.

ously embeddable in this same low-dimensional subspace. Moreover, the DE trajectories of LLMs are geometrically bounded between those of KDE and Bayesian histograms.

**5. Explaining LLaMA-2 with bespoke KDE.** We observe that LLMs' in-context DE trajectories consistently remain close to those of KDE, suggesting that LLMs might be employing a kernel-like algorithm, internally. To investigate this hypothesis, we optimize a KDE algorithm with fitted kernel parameters (using only two parameters).

We now provide a detailed explanation of how to apply InPCA to analyze in-context DE trajectories. We will also introduce two classical DE algorithms that serve as insightful comparisons.

## 3.1 InPCA visualization

To visualize the density estimation (DE) trajectories of various algorithms in a lower-dimensional space, we employ Intensive Principal Component Analysis (InPCA) (Quinn et al., 2019; Mao et al., 2024; Teoh et al., 2020). Given a set of PDFs $\{p_i\}_{i=1}^m$, InPCA aims to embed them as points $\{P_i\}_{i=1}^m$ in a lower-dimensional space $\mathbb{R}^d$ ($d \ll m$), such that the Euclidean distances between embedded points closely approximate the statistical distances between the corresponding PDFs:

$$\|P_i - P_j\|_2 \approx D(p_i, p_j), \quad \forall i, j \in \{1, \ldots, m\}, \tag{1}$$

where $D(\cdot, \cdot)$ is a chosen statistical distance measure between PDFs. In this work, we choose $D(\cdot, \cdot)$ to be the *Hellinger distance* (Hellinger, 1909), defined as:

$$D_{\text{Hel}}^2(p_i, p_j) = \frac{1}{2} \int \left| p_i(x)^{1/2} - p_j(x)^{1/2} \right|^2 \, \mathrm{d}x. \tag{2}$$

The Hellinger distance is ideal for visualizing our PDF trajectories for the following reasons[6]: 1) it locally agrees with the KL-divergence (Liese & Vajda, 2006), a non-negative measure commonly used in training modern machine learning systems, including LLMs (Touvron et al., 2023), and 2) it is symmetric and satisfies the triangle inequality, making it a proper distance metric suitable for geometric analysis (Quinn et al., 2019; Teoh et al., 2020).

Having chosen an appropriate distance measure, we proceed with InPCA involving three steps:

1. Calculate the Hellinger distance between each pair of PDFs, $D_{\text{Hel}}^2(p_i, p_j)$. This results in a pairwise distance matrix $\boldsymbol{D} \in \mathbb{R}^{m \times m}$, whose $\boldsymbol{D}_{ij}$ entry equals $D_{\text{Hel}}^2(p_i, p_j)$.

2. Obtain a "centered" matrix $W = -\frac{1}{2}L\boldsymbol{D}L$, where $L_{ij} = \delta_{ij} - 1/m$. This step is closely related to multi-dimensional scaling (MDS) (Chen et al., 2007).

3. Perform the eigenvalue decomposition $W = U\Sigma U^T$. The diagonal entries in $\Sigma$, sorted in descending order, represent the eigenvalues of $W$. The embedding coordinates for the PDFs are given by $U\Sigma^{1/2}$, where the columns of $U$ are the corresponding eigenvectors.

The eigenvalues in $\Sigma$ represent the amount of variance explained by each dimension in the embedded probability space. The cumulative fraction of total variance captured with an increasing number of dimensions is shown in Fig. 2 (c). In our main experiments, approximately 90% of the Hellinger distance variance can be captured in just two dimensions, enabling faithful 2D visualization of the DE trajectories.

## 3.2 Classical DE algorithms

**Kernel density estimation.** Kernel Density Estimation (KDE) is a non-parametric method for estimating the probability density function of a random variable based on a finite data sample. Given $n$ samples $X_1, X_2, \ldots, X_n$ drawn from some distribution with an unknown density function $f$, the kernel density estimator $\hat{p}_h(x)$ is a function over the support of $x$ defined as:

$$\hat{p}_h(x) = \frac{1}{nh} \sum_{i=1}^n K\left(\frac{x - X_i}{h}\right), \tag{3}$$

where $K$ is the kernel function and $h > 0$ is the bandwidth (smoothing parameter). The kernel function $K$ is typically chosen to be a symmetric probability density function, such as the Gaussian kernel $K(u) = \frac{1}{\sqrt{2\pi}}e^{-\frac{1}{2}u^2}$. We discuss other kernel shapes in Appendix A.7.2.

The optimal bandwidth schedule is a central object of study in classical KDE literature (Appendix A.7). By analyzing the Asymptotic Mean Integrated Squared Error (AMISE) (Equation 15), researchers have derived a widely accepted scaling for the optimal bandwidth.

$$h(n) = Cn^{-\frac{1}{5}} \tag{4}$$

---

[6]See Appendices A.5 and A.6 for viusalizations using other distance measures such as L2 and the symmetrized KL-divergence.

where $n$ is the sample size and $C$ is a pre-coefficient (Equation 16). In the low-data regime, a larger bandwidth provides more smoothing bias, which compensates for data sparsity. While the $n^{-1/5}$ scaling is widely accepted, determining the pre-coefficient $C$ is more challenging (Wand & Jones, 1994; Silverman, 2018). Unless otherwise noted, we use $C = 1$ for classical KDE in this paper[7].

**Bayesian histogram.** The Bayesian histogram (Lugosi & Nobel, 1996) is another non-parametric method for density estimation, and can be formulated as follows:

$$\hat{p}_n(x) = \frac{n_i + \alpha}{n + \alpha M}, \tag{5}$$

where $\hat{p}_n(x)$ is the estimated probability density for bin $i$ containing $x$, $n_i$ is the number of observed data points in bin $i$, $n$ is the total number of observed data points, $M$ is the total number of bins.[8], and $\alpha$ is the prior count for each bin. We set $\alpha = 1$, effectively populating each bin with one "hallucinated" data point prior to observing any data (Jeffreys, 1946). This choice ensures that the histogram algorithm starts from a state of maximal ignorance, consistent with LLaMA's in-context DE in the low data regime[9].

## 4 EXPERIMENTS AND ANALYSIS

We visualize and analyze the learning trajectories of an LLM on two types of target (ground truth) distributions: uniform and Gaussian. To provide context and facilitate interpretation of the 2D space, we embed the following additional reference points and trajectories:

- **Ignorance**: A point representing maximum entropy (uniform distribution over the entire support).
- **Truth**: A point representing the ground-truth distribution.
- **Geodesic**: The shortest trajectory connecting the Ignorance and Truth points (Mao et al., 2024).
- **Gaussian submanifold**: A 1D manifold of centered Gaussians with variances ranging from $\infty$ to 0.

For brevity, this section focuses on LLaMA-2 13B, as other LLMs yield qualitatively similar results, detailed in Appendix A.4. While LLaMA-2 has a context window of 4096 tokens (equivalent to $\sim$1365 comma-delimited, 2-digit data points), we limit our analysis to a context length of $n = 200$. This limitation is based on our observation that LLaMA's DE accuracy typically plateaus beyond this point.

### 4.1 GAUSSIAN TARGET

We begin our analysis with Gaussian target distributions of varying widths.

**Wide Gaussian target.** A wide Gaussian distribution serves as our simplest target, as it is close in Hellinger distance to the uniform distribution (total ignorance). In this scenario, both LLaMA and Gaussian KDE successfully approximate the target PDF within 200 data points.

**Narrow Gaussian target.** However, as the Gaussian target narrows, Gaussian KDE lags behind, while LLaMA maintains its ability to closely approximate the target distribution.

In all three cases, the Bayesian histogram, by design, begins exactly at maximal ignorance, and then follows the geodesic trajectory. Despite following this geometrically shortest path, the Bayesian histogram is the slowest to converge to the target distribution; likely due to the strong influence of its uniform prior.

Gaussian KDE, on the other hand, starts closer to the target, thanks to its unfair advantage of having a Gaussian-shape kernel. Interestingly, Gaussian KDE consistently lingers on the Gaussian sub-manifold throughout the DE process. This behavior of lingering on the Gaussian sub-manifold is

---

[7]In practice, various heuristics have been proposed to determine $C$, such as Silverman's rule of thumb (Silverman, 2018) (see Appendices A.7.1 and A.13).

[8]$M = 100$ since our most refined estimation from Hierarchy-PDF consists of 100 one-digit bins

[9]A curious feature which we foreshadowed in Section 3 and further discuss throughout Section 4

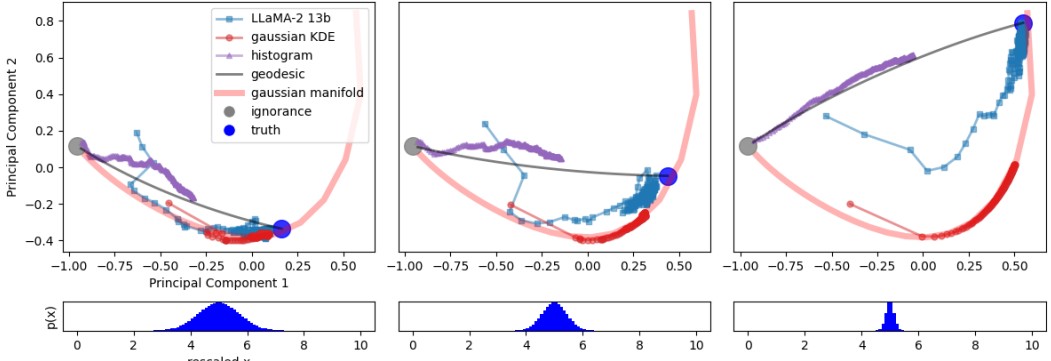

Figure 3: In-context density estimation trajectories for Gaussian targets. Top row: 2D InPCA embeddings of DE trajectories for Gaussian targets of decreasing width (left to right). Bottom row: Corresponding ground truth distributions. These 2D embeddings capture 92% of pairwise Hellinger distances between probability distributions.

not unique to Gaussian kernels; as demonstrated in Figure 16, KDEs with alternative kernel shapes (e.g., exponential and parabolic) also exhibit a strong propensity for the Gaussian submanifold. One potential explanation for this phenomenon lies in the nature of KDE itself. KDE can be expressed as a convolution of two distributions: the empirical data distribution and the kernel shape, as

$$\hat{p}_h(x) = \frac{1}{nh} \sum_{i=1}^{n} K\left(\frac{x - X_i}{h}\right) = (F_n * K_h)(x), \tag{6}$$

where $F_n$ is the empirical distribution and $K_h$ is the scaled kernel. This convolution operation tends to produce Gaussian-like distributions due to the central limit theorem (Fischer, 2011). Consequently, **a DE trajectory near the Gaussian manifold may indicate a kernel-style density estimation algorithm.**

## 4.2 UNIFORM TARGET

The Gaussian distribution easily arises in data with additive noise (Fischer, 2011) and therefore likely dominates the training data (Touvron et al., 2023). What's more, the Gaussian distribution is everywhere smooth, which makes it very easy to estimate from a function approximation point of view (DeVore, 1993). Uniform distributions, on the other hand, feature non-differentiable boundaries; difficult to represent by both parametric (DeVore, 1993) and kernel-based (Wand & Jones, 1994, Chapter 2.9) methods. For these reasons, we now investigate the in-context DE trajectory with uniform targets.

**Wide uniform target.** For a wide uniform target, both LLaMA and Gaussian KDE initially move rapidly towards the target distribution. However, they then linger at the point on the Gaussian submanifold nearest to the uniform target. As more data streams in, they slowly depart from the submanifold and converge to the target distribution.

**Narrow uniform target.** As the uniform target narrows, Gaussian KDE's performance deteriorates, while LLM-based in-context DE successfully reaches the target. Notably, the LLM exhibits less bias towards the Gaussian sub-manifold, as compared with the narrow Gaussian target case.

In the low-data regime, Gaussian KDE and Bayesian histogram follow the same trajectories that they previously followed for the narrow Gaussian target. However, the LLM already appears to differentiate this target by taking a path further from the Gaussian sub-manifold and closer to the geodesic. This behavior suggests that **LLMs' in-context DE algorithm is more flexible and adaptive than classical KDE with pre-determined width schedule and shape.**

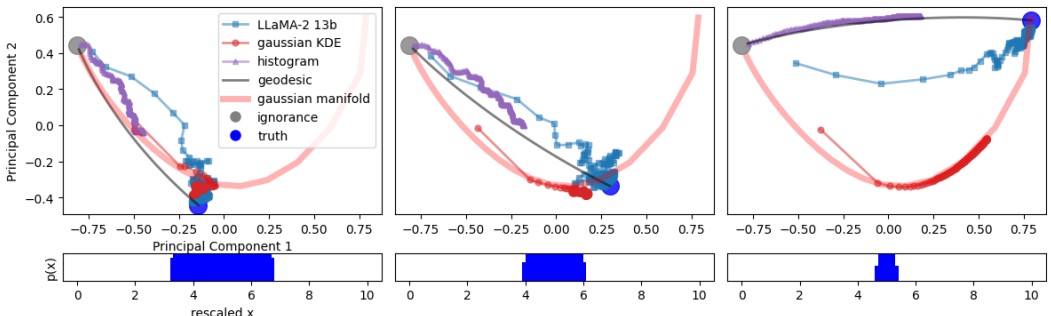

Figure 4: In-context density estimation trajectories for uniform distribution targets. Top row: 2D InPCA embeddings of DE trajectories for uniform targets of decreasing width (left to right). Bottom row: Corresponding ground truth distributions. These 2D embeddings capture 89% of pairwise Hellinger distances between probability distributions.

### 4.3 THE KERNEL INTERPRETATION OF IN-CONTEXT DENSITY ESTIMATION

Based on our observations in Sections 4.1 and 4.2, we propose a kernel-based interpretation of LLaMA's in-context density estimation algorithm. The bias towards Gaussian submanifolds in LLaMA's trajectories suggests a KDE-like approach. To test this hypothesis, we develop a bespoke KDE model with adaptive kernel shape and bandwidth, optimized to emulate LLaMA's learning trajectory.

#### 4.3.1 BESPOKE KDE MODEL

We construct our bespoke KDE model in two steps:

**Step 1: Parameterize kernel shape**

We introduce a flexible kernel function $K_s(x)$ parameterized by a shape parameter $s$:

$$K_s(x) = \frac{b(s)e^{-|b(s)x|^s}}{Z(s)} \tag{7}$$

where $Z(s) = 2\Gamma(\frac{1}{s}+1)$ normalizes the kernel to integrate to 1, and $b(s) = \sqrt{\frac{\Gamma(\frac{3}{s}+1)}{3\Gamma(\frac{1}{s}+1)}}$ scales it to maintain unit variance. This parameterization allows us to interpolate between common kernel shapes such as exponential ($s = 1$), Gaussian ($s = 2$), and tophat ($s \rightarrow \infty$), as visualized in Figure 5.

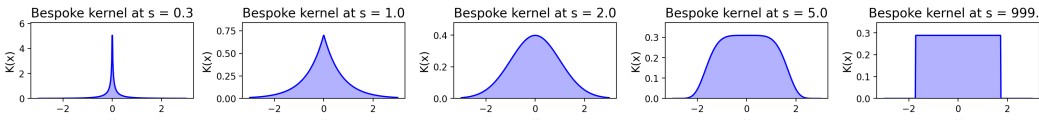

Figure 5: Bespoke kernel interpolates various common kernel shapes.

We visualize the DE trajectories with these three common kernels ($s = 1, 2$, and $\infty$) in Figure 16 of Appendix A.7.2. As the shape parameter ($s$) decreases, the DE trajectories gradually shift from outside the Gaussian submanifold to inside, moving closer to LLaMA's trajectory. This trend suggests that our parameterization (Equation 7) may be able to capture LLaMA's behavior by allowing $s$ to take values below 1, extending beyond the range of common kernel shapes.

We augment the standard KDE formula with kernel shape, resulting in two hyperparameters: $h$ and $s$.

$$\hat{p}_{h,s}(x) = \frac{1}{nh} \sum_{i=1}^{n} K_s\left(\frac{x - X_i}{h}\right) \tag{8}$$

**Step 2: Optimize kernel bandwidth (h) and shape (s)**

For a given DE trajectory $\hat{p}_1(x), \ldots, \hat{p}_n(x)$, we optimize our bespoke KDE to minimize the Hellinger distance at each context length $i$:

$$\min_{s_i \in (0,\infty), h_i \in (0,\infty)} D_{\text{Hel}}(\hat{p}_i(x) \| \hat{p}_{h_i, s_i}(x)) \tag{9}$$

yielding the "bespoke KDE" bandwidth schedule $\{h_i\}_{i=1}^n$ and shape schedule $\{s_i\}_{i=1}^n$, which together prescribe a sequence of fitted kernels of changing widths and shapes. We visualize such sequences of fitted kernels, and compare them against the Gaussian kernel with standard $n^{-1/5}$ width schedule in Appendix A.8 and A.9. We implement this optimization numerically using SciPy, and estimate parameter uncertainties from the inverse Hessian of the loss function (Kl-divergence) at the optimum. This provides error bars for our fitted kernel shape and bandwidth parameters (Figure 6). In Appendix A.12, we elaborate the information-theoretic motivation for our uncertainty quantification method.

### 4.3.2 BESPOKE KDE ANALYSIS

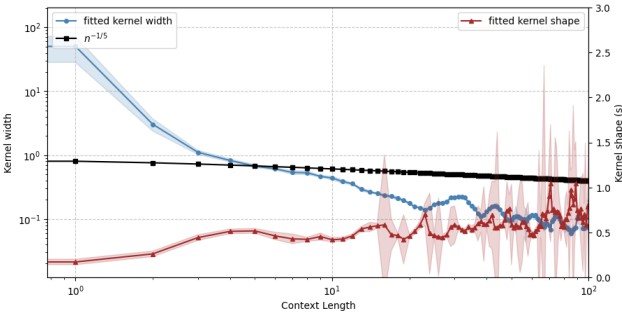

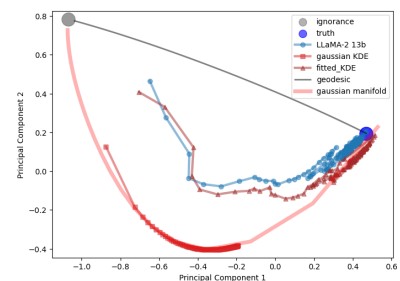

Figure 6: Optimized kernel shape ($s$) and bandwidth ($h$) schedules for the bespoke KDE model, fitted to LLaMA-2 13b's DE trajectory on a narrow Gaussian target.

Figure 7: Bespoke KDE trajectories ($\circ$) fitted to LLaMA-2 13b's DE trajectory ($\triangle$) on a narrow Gaussian target.

Figure 6 presents the optimized kernel shape and bandwidth schedules for our bespoke KDE model, fitted to LLaMA-2 13b's in-context DE trajectory for the narrow Gaussian target (Section 4.1). Two key observations emerge:

1. The fitted kernel width decays significantly faster than the standard KDE bandwidth schedule ($n^{-\frac{1}{5}}$). This rapid decay explains the much longer trajectory of LLaMA's in-context DE within the inPCA embedding, as compared to standard Gaussian KDE (Figure 3).

2. Unlike fixed-shape KDE methods (e.g., Gaussian KDE with $s = 2$), LLaMA seems to implicitly employ an adaptive kernel. The shape parameter $s$ evolves from $\sim 0.1$ to $\sim 1$, with increasing uncertainty for $n \gtrsim 20$.

Figure 7 compares LLaMA-2 13b's in-context DE trajectory with its bespoke KDE counterpart, demonstrating the close fit achieved by our model.

**Significance of Bespoke KDE** The existence of a bespoke KDE that closely imitates an LLM's in-context DE processs is significant and non-trivial. Not all DE processes can be faithfully represented by KDE methods; for instance, the Bayesian histogram, despite its superficial similarity to a fixed-width tophat kernel KDE, resists accurate modeling as a bespoke KDE process (Appendix A.3).

While our initial experiments focused on regular distributions like Gaussian and uniform, for better generality we extended our experiments to randomly generated PDFs (Appendix A.9). Notably, the LLMs' DE performance on these irregular distributions can still be effectively described using our bespoke KDE process (Appendix A.8 and A.9). Although the DE trajectories of these random PDFs lack low-dimensional embeddings, and thus provide less geometric insight, they nonetheless corroborate our findings from the cases with more regular distributions.

This consistency across diverse target PDF types strengthens our core hypothesis about the in-context DE mechanism underlying LLMs. Specifically, we posit that an LLM's in-context DE algorithm shares fundamental characteristics with kernel density estimation, albeit with adaptive kernel shape and bandwidth schedules that distinguish it from classical KDE approaches.

## 5 CONCLUSIONS

Inspired by emergent, in-context abilities of LLMs to continue stochastic time series, the current work explores the efficacy of foundation models operating as kernel density estimators; a crucial element within such time series forecasting. Through an application of InPCA, we determine that such in-context kernel density estimation proceeds within a common, low-dimensional probability space; along meaningful trajectories that allow for a comparison between histogram, Gaussian kernel density estimation, and LLM in-context kernel density estimation. Through the lens of InPCA, it becomes clear that a profitable characterization of this in-context learning can be made in terms of a two-parameter, adaptive kernel density estimation framework; hinting at mechanistic basis (Appendix A.1) that points in the direction of future research.

**Future direction: towards dispersive induction heads**. Recent research has identified induction heads as fundamental circuits underlying many in-context learning behaviors in discrete-state stochastic systems (Olsson et al., 2022; Bigelow et al., 2024). These emergent circuits increase the predicted probability of token combinations that are observed in-context. However, such discrete mechanisms are insufficient to explain the in-context learning of continuous stochastic systems, such as the density estimation tasks we have studied. To address this gap, our Bespoke KDE analysis in Section 4.3.2 reveals that LLMs might possess a kernel-like induction mechanism, which we term **dispersive induction head**. This is an extension to the induction head concept, and operates as follows:

- Unlike standard induction heads (Akyürek et al., 2024), a dispersive induction head increases the predicted probability of not only exact matches, but also similar tokens or words.
- The "similarity" is determined by an adaptive kernel, analogous to our bespoke KDE model.
- The influence of each observation on dissimilar tokens decays over context length, mirroring the decreasing bandwidth in our KDE model.

This concept of dispersive induction heads could potentially bridge the gap between discrete (Akyürek et al., 2024) and continuous (Gruver et al., 2024; Liu et al., 2024) in-context learning mechanisms in transformers (Dong et al., 2024).

### ACKNOWLEDGEMENTS

This work was supported by the SciAI Center, and funded by the Office of Naval Research (ONR), under Grant Numbers N00014-23-1-2729 and N00014-23-1-2716.

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

# A APPENDIX

## A.1 MECHANISTIC INTERPRETATIONS OF IN-CONTEXT LEARNING.

In recent years a paradigm has emerged for LLMs' in-context learning abilities: the simulation and application of smaller, classical models, with well-understood algorithmic features, in response to prompt information. As an example, Olsson et al. (2022) identified "induction heads" in pre-trained LLMs, which simulate 1-gram models for language learning. Then, Akyürek et al. (2024) extended this analysis to "n-gram heads" for computing distributions of next tokens conditioned on n previous tokens. It has also been observed in the literature that transformers can perform in-context regression via gradient descent (von Oswald et al., 2023; Dai et al., 2023). These works proposed to understand induction heads as a specific case of in-context gradient descent. Akyürek et al. (2023) showed that trained in-context learners closely match predictions of gradient-based optimization algorithms. More recently, Kantamneni et al. (2024) identified mechanisms in transformers that implicitly implement matrix exponential methods for solving linear ordinary differential equations (ODEs).

Our approach differs from these aforementioned studies for the following reason: we do not train LLMs on synthetic data designed to induce specific in-context learning abilities. Instead, we investigate the emergent density estimation capabilities of pre-trained foundation models (the LLaMA-2 suite) without any fine-tuning. This approach aligns more closely with recent works (Liu et al., 2024; Gruver et al., 2024) that focus on the inherent mathematical abilities of foundations models, rather than LLMs that are trained to induce certain behaviors.

## A.2 LOW-DIMENSIONAL STRUCTURES IN LEARNING DYNAMICS.

Despite the success of modern neural networks at learning patterns in high-dimensional data, the learning dynamics of these complex neural networks are often shown to be constrained to low-dimensional, potentially non-linear subspaces. Hu et al. (2021) demonstrated that the fine-tuning dynamics of LLMs such as GPT and RoBERTa can be well-captured within extremely low-rank, weighted spaces. More recently, Mao et al. (2024) showed that, during training, neural networks spanning a wide range of architectures and sizes trace out similar low-dimensional trajectories in the space of probability distributions. Their work focuses on learning trajectories $p_{k=0}, ..., p_{k=t}$, where $k$ indexes the training epoch, and $p_k$ is a high-dimensional PDF describing the model's probabilistic classification of input data. Key to their observations is a technique termed *Intensive Principal Component Analysis (InPCA)*, a visualization tool (Quinn et al., 2019) that embeds PDFs as points in low dimensional spaces, such that the geometric (Euclidean or Minkowski) distances between embedded points reflect the statistical distances - e.g. the Hellinger or Bhattacharyya distance - between the corresponding PDFs.

Inspired by Mao et al. (2024) and Quinn et al. (2019), we extend this line of inquiry to investigate whether the *in-context* learning dynamics of LLMs also follow low-dimensional trajectories in probability space. and $\hat{p}_k$ is a PDF of dimension $10^N$, where N is the number of digits used in the multi-digit representation (see 3). Each $\hat{p}_k$ describes the model's estimation of the underlying data distribution at a certain context length. Our findings show that in-context density estimation traces low-dimensional paths.

## A.3 BESPOKE KDE CANNOT IMITATE ALL DE PROCESSES

This section examines the ability of Bespoke KDE to imitate various density estimation (DE) processes, including LLaMA-2 models (7b, 13b, 70b), Bayesian histogram, and standard Gaussian KDE.

### A.3.1 DE TRAJECTORIES OF LLaMA-2 7B, 13B, AND 70BF VS. BESPOKE KDE IMITATIONS

Figure 8 illustrates the DE trajectories of various models alongside their bespoke KDE counterparts. We observe that: 1. Bespoke KDE successfully imitates the LLaMA suite with high precision. 2. It trivially replicates the Gaussian KDE trajectory, as expected. 3. However, it struggles to accurately capture the Bayesian histogram's trajectory. To further investigate these observations, we analyze the fitted kernel parameters for Gaussian KDE and Bayesian histogram:

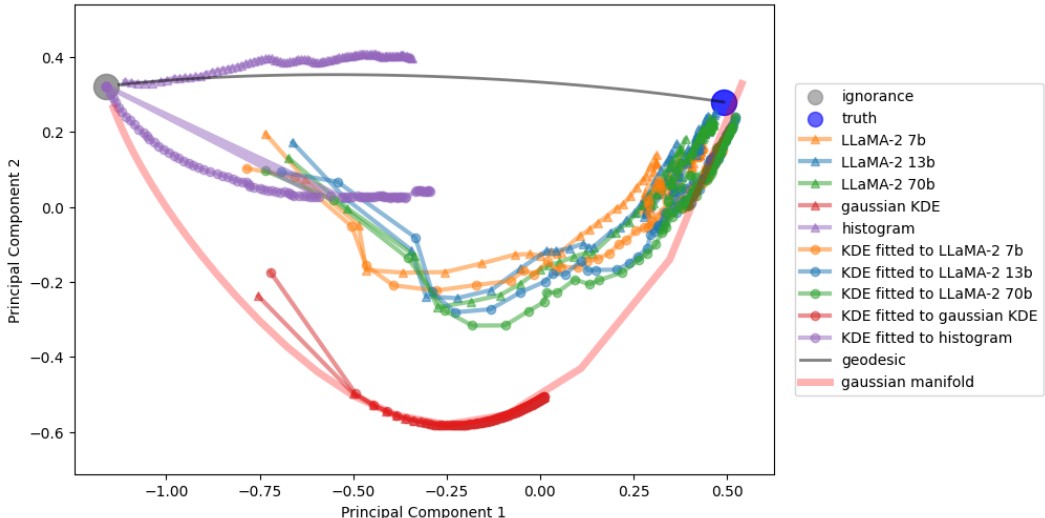

Figure 8: Comparison of original DE trajectories (△) and their bespoke KDE imitations (○) for LLaMA-2 models (7b, 13b, 70b), Gaussian KDE, and Bayesian histogram. The 2D embeddings capture 97% of pairwise Hellinger distances between probability distributions.

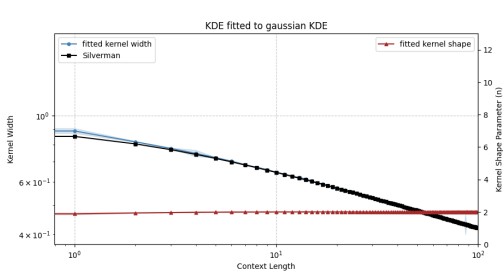

Figure 9: Bespoke KDE parameters fitted to standard Gaussian-kernel KDE. The model accurately recovers the Gaussian kernel shape ($s = 2$) and bandwidth schedule ($h = n^{-\frac{1}{5}}$).

Figure 10: Bespoke KDE parameters fitted to Bayesian histogram. The constant fitted kernel width likely reflects the fixed bin width, while the near-zero shape parameter indicates a highly peaked distribution. The high uncertainties suggest fundamental mismatch between the two models.

Figure 9 demonstrates that bespoke KDE accurately recovers the parameters of standard Gaussian KDE. In contrast, Figure 10 reveals significant challenges in fitting Bespoke KDE to the Bayesian histogram: 1. The fitted kernel width remains constant, likely reflecting the fixed bin width of the histogram. 2. The near-zero shape parameter suggests a highly peaked distribution. 3. High uncertainties in both parameters indicate a fundamental mismatch between Histogram and KDE.

These results highlight that while Bespoke KDE can effectively model certain DE processes (e.g., LLMs and Gaussian KDE), it cannot capture fundamentally different approaches like the Bayesian histogram.

### A.3.2 Meta-InPCA Embedding of Trajectories: the LLaMA-2 suite

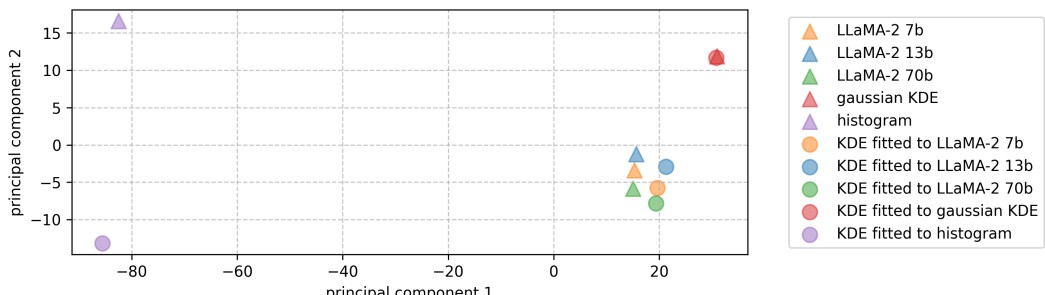

Figure 11: Meta-InPCA embedding of DE trajectories ($\triangle$) and their bespoke KDE imitations ($\circ$). This 2D embedding captures 94% of pairwise meta-distances between trajectories.

To quantify the similarity between the bespoke KDE model and various DE processes, we introduce a meta-distance measure for trajectories and apply InPCA at a higher level of abstraction.

**Trajectory distance metric.** We define a meta-distance between two trajectories $\{p_i\}_{i=1}^{n}$ and $\{q_i\}_{i=1}^{n}$ as the sum of Hellinger distances between corresponding points at each context length:

$$D_{traj}(\{p_i\}, \{q_i\}) = \sum_{i=1}^{n} D_{\text{Hel}}(p_i, q_i) \tag{10}$$

where $D_{\text{Hel}}$ is the Hellinger distance defined in Equation 2.

**Meta-InPCA procedure.** Given a set of trajectories $\text{traj}_1, ..., \text{traj}_l$, we: 1. Compute the pairwise trajectory distance matrix $D \in \mathbb{R}^{l \times l}$ using Equation 10. 2. Apply the InPCA procedure described in Section 3.1 to embed these trajectories in a lower-dimensional space.

Unlike previous InPCA visualizations where each point represented a single PDF, in this meta-embedding, each point represents an entire DE trajectory.

**Observations.** Figure 11 reveals several key insights: 1. The Bayesian histogram and its bespoke KDE imitation are far apart, confirming the model's inability to capture this approach. 2. Gaussian KDE almost exactly overlaps with its bespoke KDE imitation, as expected. 3. LLaMA-2 models (7b, 13b, 70b) are relatively close to their bespoke KDE counterparts but do not overlap exactly. These observations suggest that while the 2-parameter bespoke KDE model captures much of LLaMA-2's in-context DE behavior, there are still certain nuances in LLaMA-2's algorithm that it doesn't fully encapsulate.

## A.4 META-INPCA EMBEDDING OF TRAJECTORIES: LLAMA, GEMMA, AND MISTRAL

This section documents additional experiments on three other recently released LLMs: Gemma-2b and -7b (GemmaTeam et al., 2024) and Mistral-7b-v0.3 (Jiang et al., 2023). These models share similar tokenizers as LLaMA-2, and work well with the Hierarchy-PDF algorithm Liu et al. (2024).

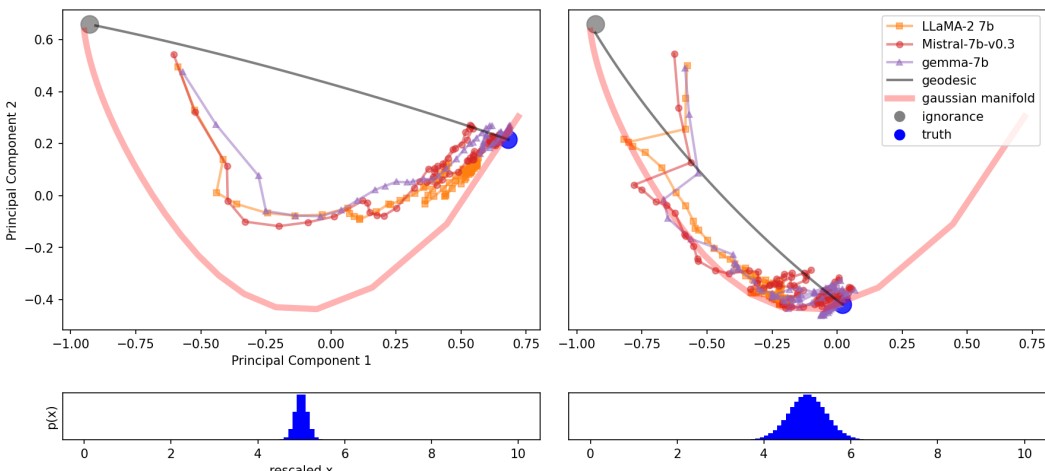

Figure 12: Comparison of in-context density estimation trajectories for LLaMA, Gemma, and Mistral. Top row: 2D InPCA embeddings of DE trajectories for narrow and wide Gaussian targets. Bottom row: Corresponding ground truth distributions. These 2D embeddings capture 91% of pairwise Hellinger distances between probability distributions.

As shown in Figure 12, the in-context DE trajectories of LLaMA-2, Mistral v0.3, and Gemma, are strikingly similar, despite the fact that they were built and trained entirely independently by different teams.

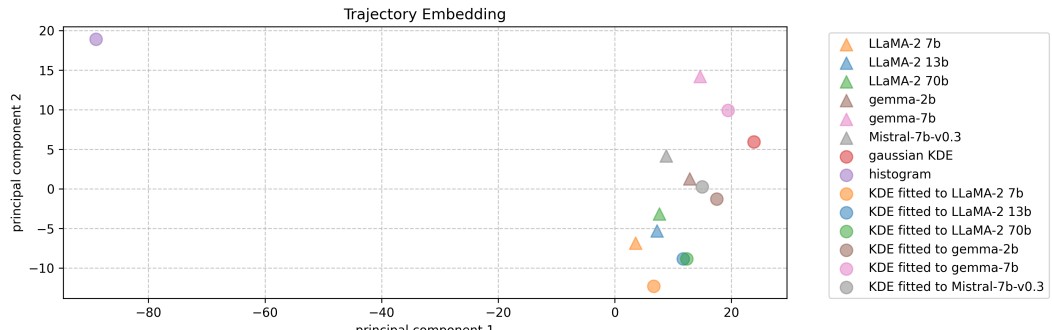

Figure 13: Meta-InPCA embedding of DE trajectories (△) and their bespoke KDE imitations (○). This 2D embedding captures 91% of pairwise meta-distances between trajectories.

To quantify the similarity of these models' learning trajectories, we use the meta-InPCA embedding introduced in Appendix A.3.2 to visualize the pairwise trajectory distance from these models. As shown in Figure 13, Gemma and Mistral are well-approximated by bespoke KDE counterparts. In fact, all kernel-like methods are quite similar to each other, forming a cluster on the right corner of the embedding, with Bayesian histogram isolated on the left, similar to Figure 11.

Interestingly, Gemma-2b seems more similar to Mistral-7b-v0.3 and LlaMA-2 70b, than to Gemma-7b. Models from the same suites are not always the most alike in terms of in-context DE trajectories.

To summarize, the kernel-like density estimation process observed in this work is not limited to any specific suites of LLMs. We therefore speculate that LLMs might spontaneously converge to a universal in-context DE algorithm.

A.5   APPENDIX: COMPARISON OF INPCA AND STANDARD PCA

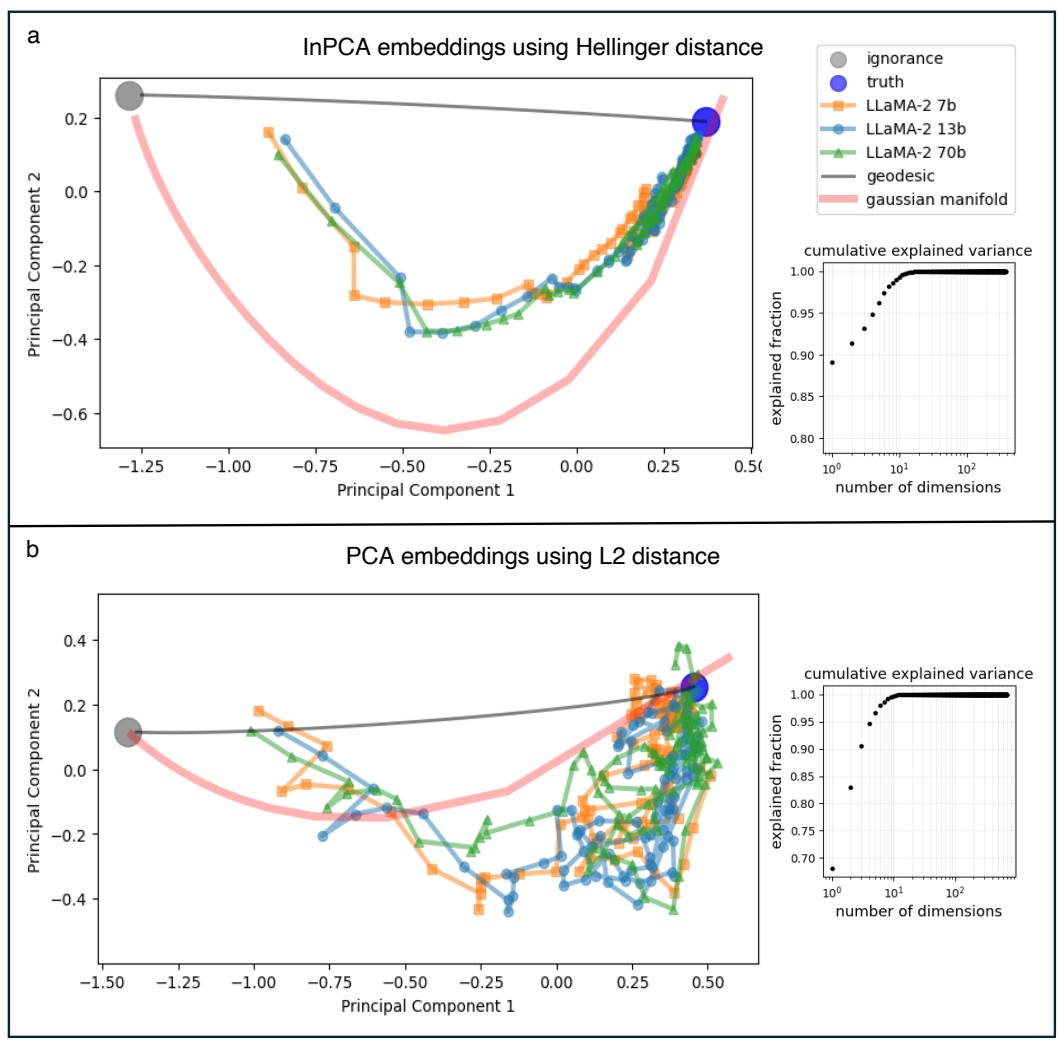

Figure 14: Comparison of InPCA and PCA visualizations for density estimation trajectories of LLaMA-2 models. (a) 2D InPCA embedding preserves 91% of pairwise Hellinger distances, revealing LLaMA's DE trajectories geometrically bounded between the geodesic and Gaussian submanifold. (b) In contrast, 2D PCA preserves only 83% of pairwise L2 distances, displaying erratic oscillations in LLaMA's DE trajectories without clear geometric relationships.

## A.6 INPCA EMBEDDINGS WITH SYMMETRIZED KL-DIVERGENCE AND BHATTACHARYYA DISTANCE

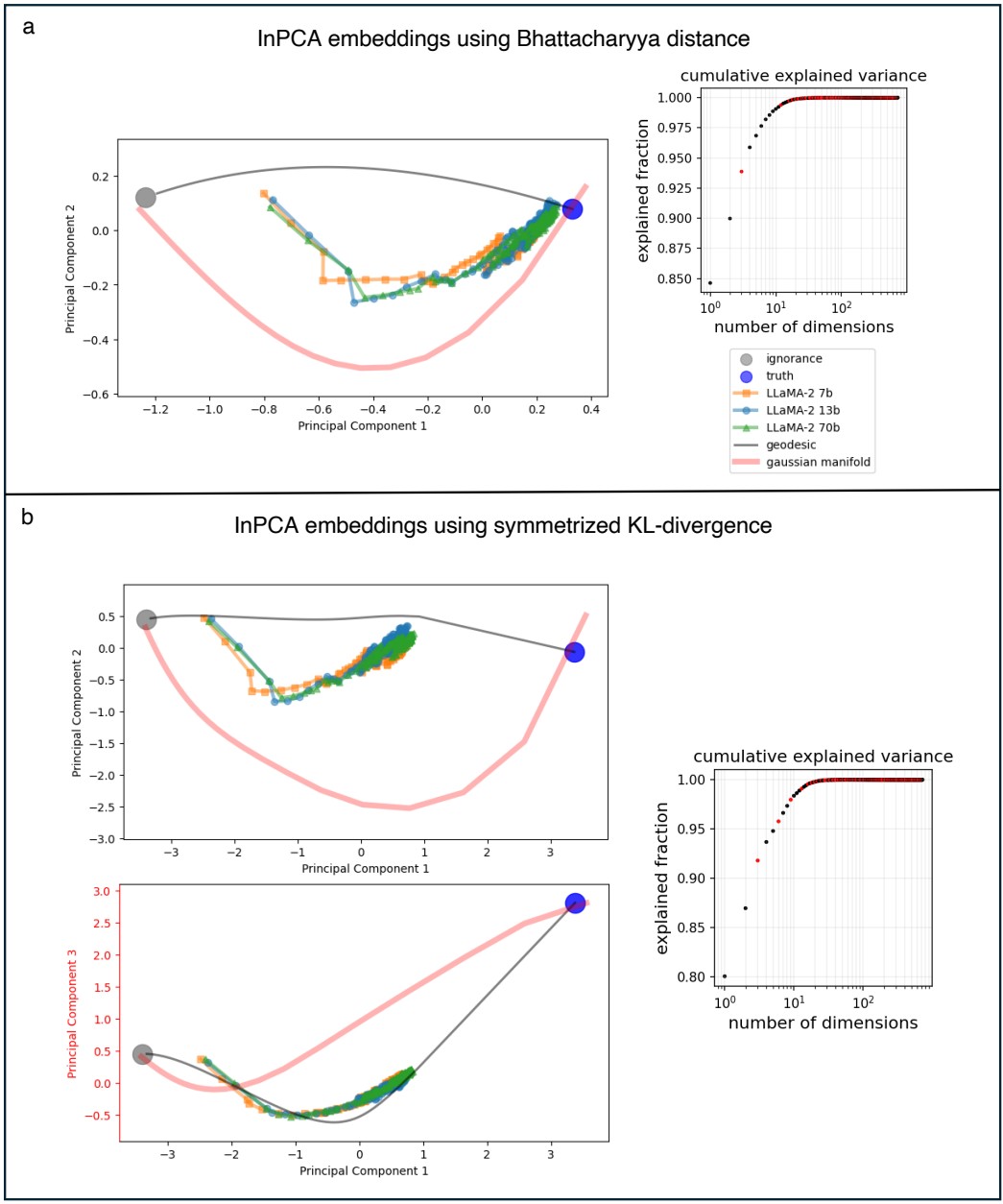

Figure 15: InPCA embeddings of LLaMA-2 in-context DE trajectories using symmetrized KL-divergence and Bhattacharyya distance. The target function is narrow Gaussian distribution as shown in Figure 2. Red dots denote negative eigenvalue, and red axis denotes imaginary principal component. (a) 2D InPCA embedding preserves 90% of pairwise Bhattacharyya distances. (b) 3D InPCA embedding preserves 87% of pairwise symmetrized KL-divergence.

Bhattacharyya distances (Quinn et al., 2019) symmetrized KL-divergence (Teoh et al., 2020) are two other statistical divergences commonly used for InPCA embeddings. They are defined as:

$$D_{\text{BT}}^2(p, q) = -\ln\left(\int \sqrt{p(x)q(x)}\,\mathrm{d}x\right) \tag{11}$$

$$D_{\text{sKL}}^2(p, q) = \int p(x)\ln\frac{p(x)}{q(x)}\,\mathrm{d}x + \int q(x)\ln\frac{q(x)}{p(x)}\,\mathrm{d}x \tag{12}$$

Figure 15 shows the InPCA embeddings resulting from these two distances. Both of these distances locally agrees with the Hellinger distance as well as the KL-divergence (Teoh et al., 2020). In fact, the Bhattacharyya distance has a deep physical connection to Hellinger distance, which was discussed in Quinn et al. (2019).

However, unlike the Hellinger distance used in this work, both Bhattacharyya distance and symmetrized KL-divergence violate the triangle in-equality (Quinn et al., 2019), and are therefore not valid distance metrics. Consequently, the eigenvalue decomposition step (Section 3.1) might result in negative eigenvalues, with the corresponding coordinates being imaginary. A linear space with a mixture of real and imaginary axes is a generalization of the Minkowski space, which is useful in special relativity (Einstein, 1905), where events are represented as points in spacetime with one timelike (imaginary) dimension, and up to three spacelike (real) dimensions. The Minkowski metric between two points $p = (x, t)$ and $q = (x', t')$ is given by:

$$d^2(p, q) = (x - x')^2 - (t - t')^2 \tag{13}$$

In our visualization context, the imaginary axis (colored red in Figure 15) plays a role similar to the time dimension in Minkowski space. **The further separated two points are along the imaginary axis, the closer they are in real distance.**

As shown in Panel (a), visualization using the Bhattacharyya distance is quantitatively the same as its Hellinger counterpart, shown in Figure 2. Visualization with symmetrized KL-divergence, shown in the top figure in Panel (b), is also qualitatively similar, except that here LLaMA's estimated density no longer seems to converge to the ground truth (blue dot). However, this is a false impression. The bottom figure in Panel (b) reveals a third principal component which is imaginary. The end point of LLaMA's estimated density is separated from the ground truth (blue dot) along both the real and imaginary axes, by about the same distance. The positive and negative distances from spacelike and timelike seprations roughly cancel out, which means the statistical distance between LLaMA's estimated density and the ground truth is actually small.

Negative eigenvalues from these metrics might be physically meaningful in certain circumstance, such as shown in Teoh et al. (2020); Quinn et al. (2019); Mao et al. (2024). However, for the purpose of visualizing in-context DE trajectories, they complicate the geometric analysis without adding much insight.

## A.7 Additional notes on Kernel Density Estimation

This section provides more background on classical results regarding optimal bandwidth schedules $\{h_i\}_{i=1}^n$ and kernel shapes. These results are derived by minimizing the Mean Integrated Squared Error (MISE) (Wand & Jones, 1994, Chapter 2.3):

$$\text{MISE}(p|\hat{p}_h) = \mathbb{E}_{x \sim p} \left[ \int (\hat{p}_h(x) - p(x))^2 \, dx \right],$$
(14)

where $\hat{p}_h$ is the kernel density estimate with bandwidth $h$, and $p$ is the true density.

### A.7.1 Optimal Bandwidth

While MISE (Equation 14) provides a reasonable measure of estimation error, it is often analytically intractable. To overcome this limitation, researchers developed the Asymptotic Mean Integrated Squared Error (AMISE) (Wand & Jones, 1994, Chapter 2.4), an approximation of MISE that becomes increasingly accurate as the sample size grows large:

$$\text{AMISE}(h) = \frac{1}{nh} R(K) + \frac{1}{4} h^4 \mu_2(K)^2 R(p''),$$
(15)

where $R(f) = \int f(x)^2 \, dx$, $\mu_2(K) = \int x^2 K(x) \, dx$, and $p''$ is the second derivative of the true density.

AMISE provides a more manageable form for mathematical analysis. By minimizing AMISE with respect to $h$, we can derive the optimal bandwidth:

$$h_{opt} = \left( \frac{R(K)}{n \mu_2(K)^2 R(p'')} \right)^{1/5} = C n^{-\frac{1}{5}}.$$
(16)

It's important to note that this expression involves many unknown quantities related to the true density $p$, such as the average curvature $R(p'')$, which is not known a priori. The key insight from this derivation is therefore the $n^{-1/5}$ scaling of the optimal bandwidth. In practice, there are many methods for estimating the pre-coefficient $C$ from data, such as Silverman's rule of thumb (Silverman, 2018):

$$h = 0.9 \min(\hat{\sigma}, \frac{\text{IQR}}{1.34}) n^{-\frac{1}{5}},$$
(17)

where $\hat{\sigma}$ is the sample standard deviation, IQR is the interquartile range, and $n$ is the sample size. This rule is derived heuristically based on minimizing the AMISE (Equation 15). The $n^{-1/5}$ scaling reveals a fundamental trade-off in kernel density estimation: as more data becomes available, we can afford to use a narrower kernel, but the rate at which we can narrow the kernel is relatively slow.

### A.7.2 Optimal kernel shape

The kernel function $K$ can take various forms. Common choices include:

- Gaussian kernel: $K(u) = \frac{1}{\sqrt{2\pi}} e^{-\frac{1}{2} u^2}$

- Exponential kernel: $K(u) = \frac{1}{2} e^{-|u|}$

- Epanechnikov kernel: $K(u) = \frac{3}{4}(1 - u^2)$ for $|u| \leq 1$, 0 otherwise

- Tophat kernel: $K(u) = \begin{cases} \frac{1}{2} & \text{for } |u| \leq 1 \\ 0 & \text{otherwise} \end{cases}$

The Epanechnikov kernel is a key result from optimal kernel theory as it minimizes the MISE.

As shown in Figure 16, different kernel shapes can lead to varying density estimation trajectories. This visualization compares the performance of LLaMA-13b with various kernel density estimators in a 2D InPCA embedding, capturing 87% of the pairwise Hellinger distances between probability distributions.

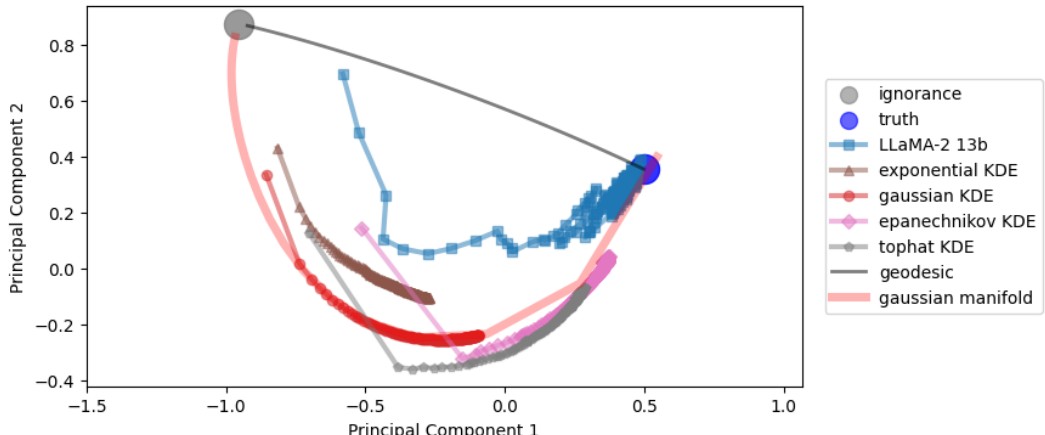

Figure 16: 2D InPCA embedding of density estimation trajectories for LLaMA-13b and KDEs with various kernel shapes: exponential ($s = 1$), Gaussian ($s = 2$), and tophat ($s = \infty$). As $s$ decreases, the DE trajectories gradually shift from outside the Gaussian submanifold to inside, moving closer to LLaMA's trajectory. The ground truth is a narrow Gaussian distribution. This visualization captures 87% of the pairwise Hellinger distances between probability distributions.

### A.7.3  MISE VS. VALID STATISTICAL DISTANCES

In our InPCA analysis, we choose the the Hellinger distance because: **1.** It locally agrees with the KL-divergence (Liese & Vajda, 2006); a measure motivated by information theory and commonly used in training modern machine learning systems, including LLMs (Touvron et al., 2023). Mathematically, this agreement means for distributions $p$ and $q$ that are close: $D^2_{Hel}(p, q) \approx \frac{1}{2}\text{KL}(p||q)$ **2.** Unlike the closely-related Bhattacharyya distance (Bhattacharyya, 1943), KL-divergence, or cross-entropy (Thomas & Joy, 2006), Hellinger distance is both symmetric and satisfies the triangle inequality, making it a proper metric. This property makes it more suitable for visualization purposes.[10] **3.** Although the $L_2$ distance $||p - q||^2_2 = \int |p(x) - q(x)|^2 \mathrm{d}x$ is also a proper distance metric, it does not locally agree with any information-theoretic divergence measure (Amari, 2016) and is therefore not suited to measure distance between PDFs.[11]

In contrast, classical KDE theory focuses on minimizing the Mean Integrated Squared Error (MISE), which is fundamentally an $L_2$-type distance that disagrees (globally and locally) with information-theoretic divergence measure. Nevertheless MISE is widely used in classical KDE literature for several reasons:

1. It can be easily analyzed mathematically using bias-variance decompositions (Wand & Jones, 1994, Chapter 2.3).

2. It leads to closed-form solutions for optimal kernels and bandwidths under certain assumptions.

3. It provides a tractable objective function for theoretical analysis and optimization.

As a consequence of its popularity and mathematical tractability, MISE has been used to derive heuristic bandwidth schedules, such as Silverman's rule (Equation 17).

We note that modern machine learning often prefers loss functions from the $f$-divergence family (Rényi, 1961), such as KL-divergence, cross-entropy, and Hellinger distance (Equation 2). These measures have strong information-theoretic motivations and are often more appropriate for probabilistic models.

---

[10]InPCA embedding with Bhattacharyya distance or symmetrized KL-divergence typically results in negative distances in Minkowski space, which are harder to interpret (Teoh et al., 2020)

[11]Consequently, PCA embedding with such naive distance measure results in erratic trajectories with obscure geometrical features, as explored in Appendix 14

Despite the prevalence of $f$-divergences in machine learning, to the best of our knowledge, there are no rigorous derivations of optimal kernel shapes or bandwidth schedules based on these measures. This gap presents an interesting avenue for future research, potentially bridging classical statistical theory with modern machine learning practices. Bridging this gap is beyond the scope of this paper. However, we speculate that this gap is the reason why the optimal kernel shape employed by LLMs differs significantly from those in classical optimal kernel theory (namely, Gaussian and Epanechnikov).

## A.8 ADAPTIVE KERNEL VISUALIZATION: COMPARING LLAMA-2 WITH KDE

This section provides a comprehensive visual analysis of DE processes across various target distributions, including Gaussian, uniform, and randomly generated probability density functions.

We present side-by-side comparisons of the DE trajectories for LLaMA-2, Gaussian KDE, and KDE with fitted kernel designed to emulate LLaMA-2's behavior. The fitted KDE process closely mirrors LLaMA-2's estimation patterns, while both diverge significantly from traditional Gaussian KDE approaches.

**Narrow Gaussian distribution target**

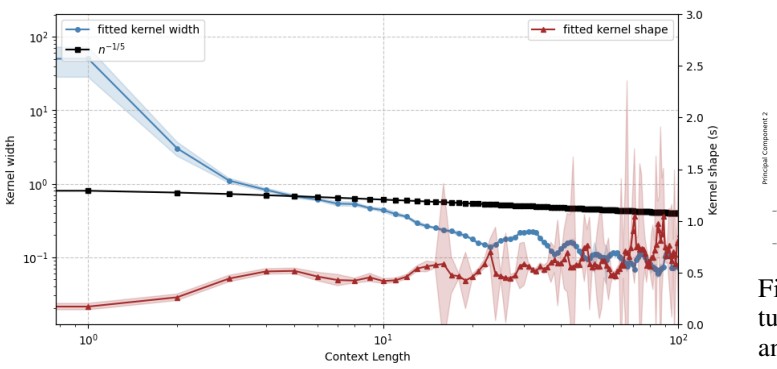

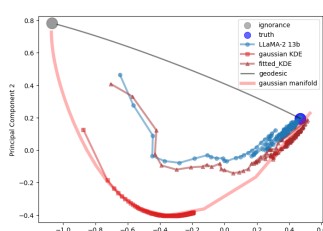

Figure 18: 2D inPCA capturing 94% of Hellinger variances

Figure 17: Fitted kernel width and shape schedule

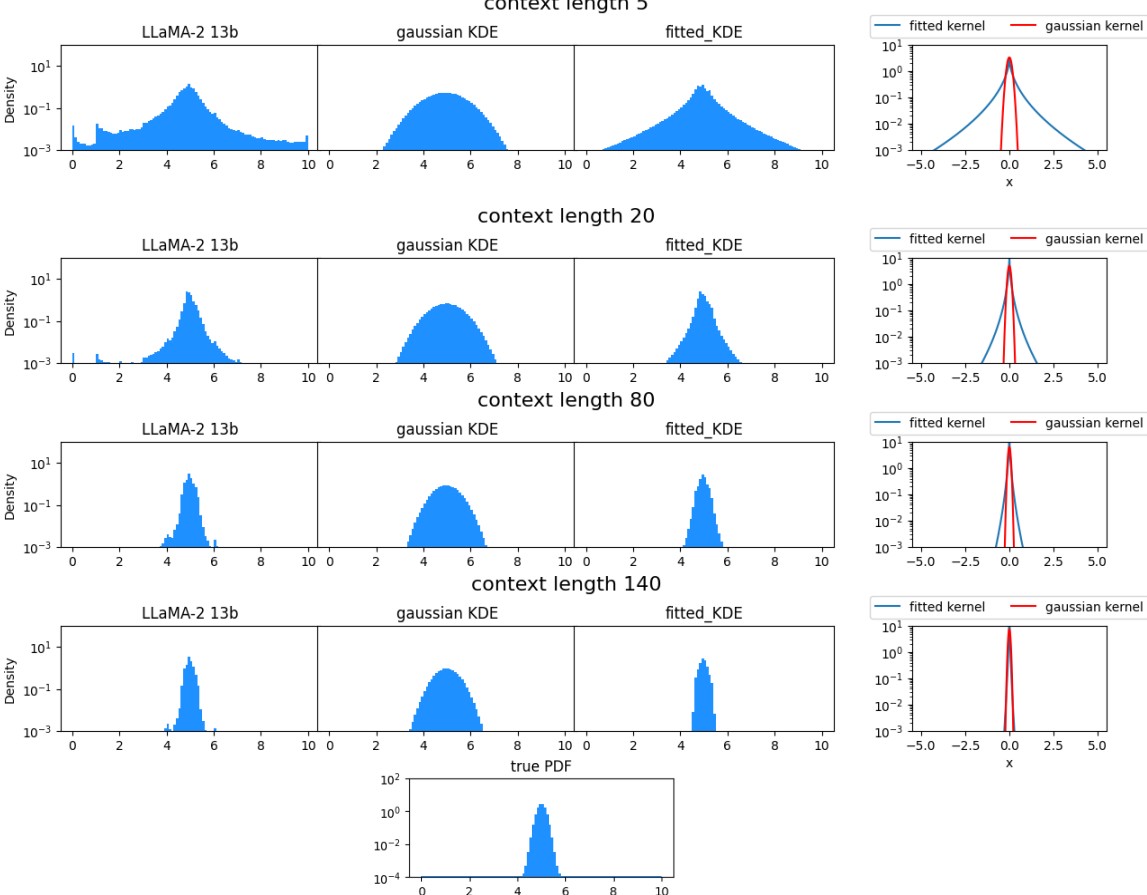

Figure 19: Fitted kernel visualization with narrow gaussian target

**Wide Gaussian distribution target**

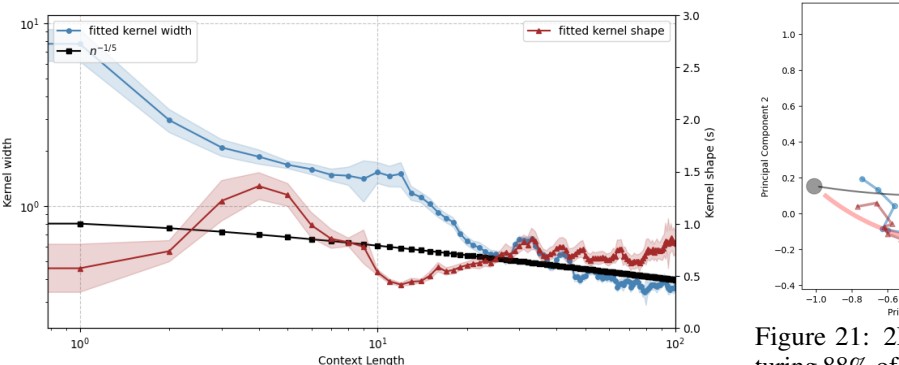

Figure 20: Fitted kernel width and shape schedule

Figure 21: 2D inPCA capturing 88% of Hellinger variances

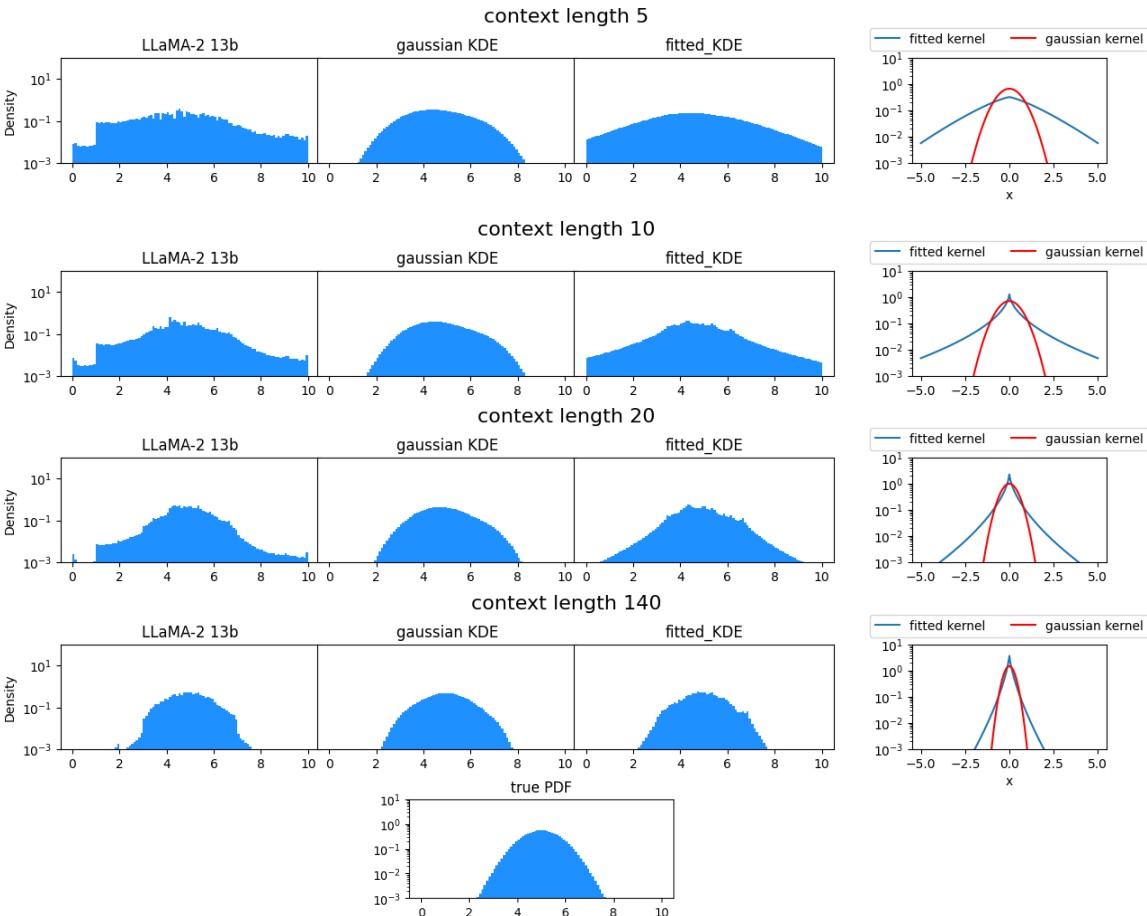

Figure 22: Fitted kernel visualization with wide gaussian target

**Narrow uniform distribution target**

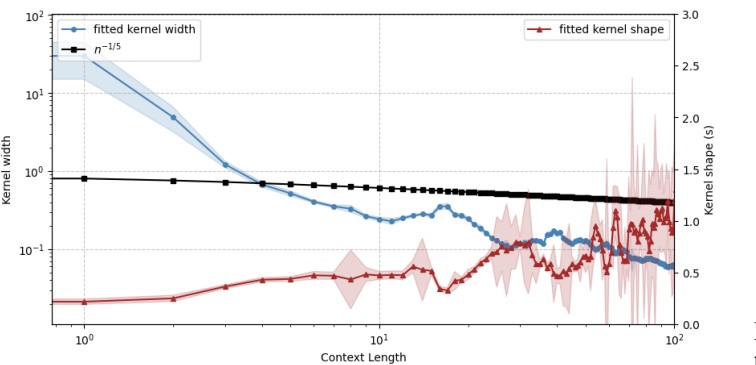

Figure 23: Fitted kernel width and shape schedule

Figure 24: 2D inPCA capturing 92% of Hellinger variances

Figure 25: Fitted kernel visualization with narrow uniform target

**Wide uniform distribution target**

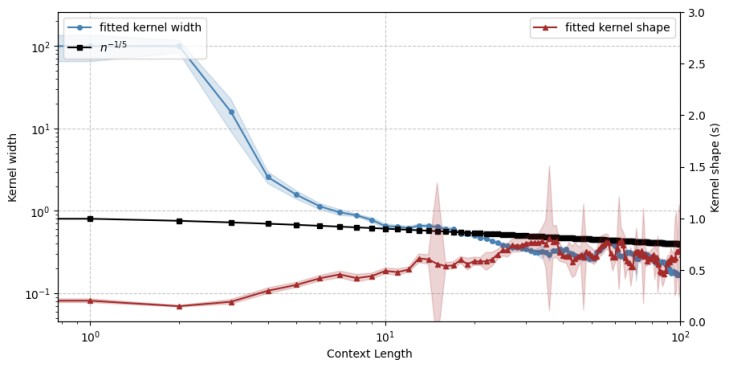

Figure 26: Fitted kernel width and shape schedule

Figure 27: 2D inPCA capturing 83% of Hellinger variances

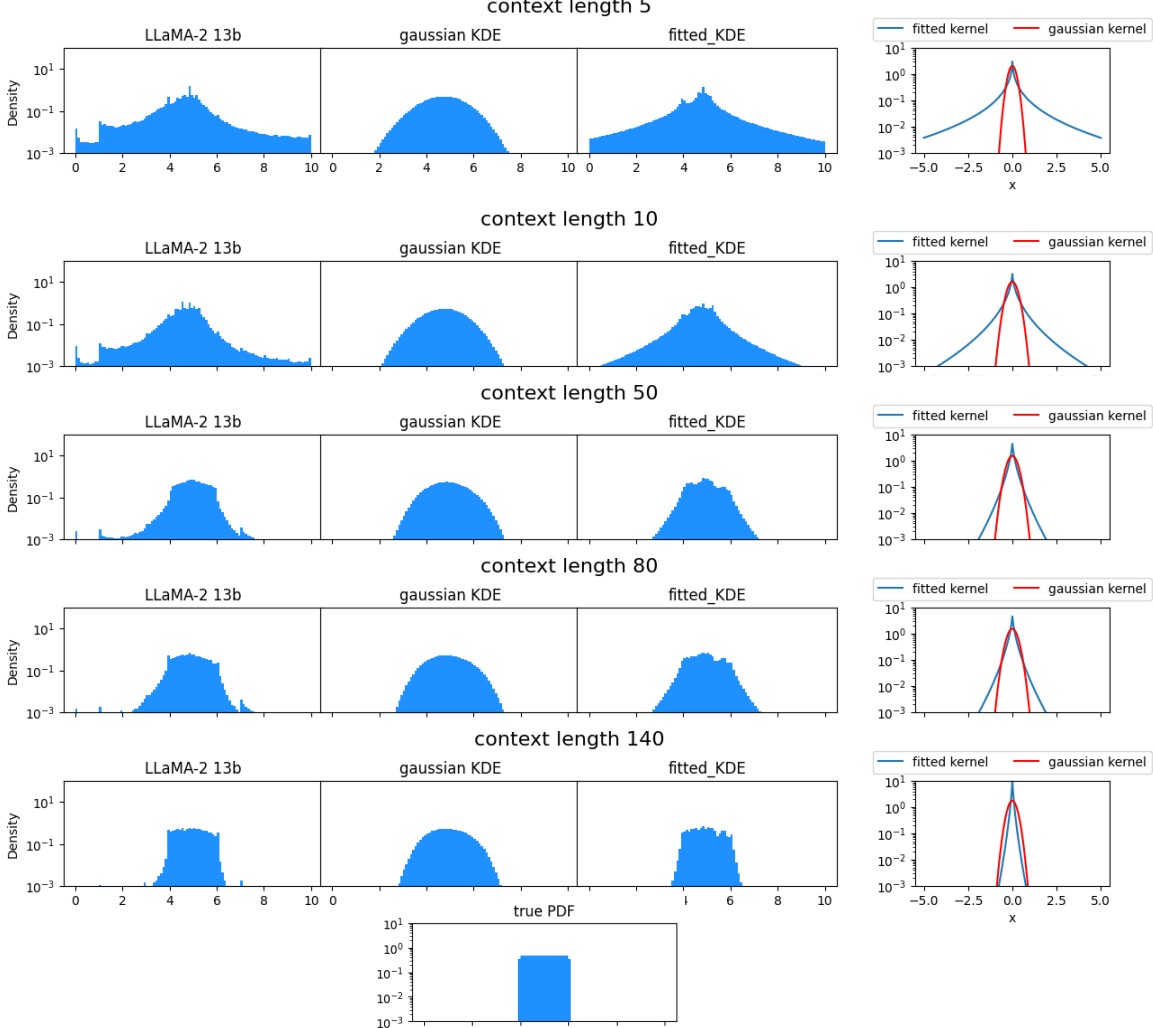

Figure 28: Fitted kernel visualization with wide uniform target

## A.9 RANDOMLY GENERATED PDFS

For completion, we also test LLaMA-2's in-context DE ability on randomly generated PDFs. As noted in Appendix A.7 and Chapter 2.9 of Wand & Jones (1994), PDFs with high average curvature (defined as $R(p'') = \int p''(x)^2 dx$) are more difficult to estimate from data. Therefore, we want to be able to control the average curvature $R(p'')$ of the generated random PDFs. We employ a technique for generating random PDF using Gaussian processes. This allows us to create a wide variety of smooth, continuous distributions with controllable average curvatures.

**Gaussian Process Generation:** We generate random PDFs using a Gaussian process with a predefined covariance matrix. The process is defined over the interval [0, 1], discretized into $N_x = 10^p$ points, where $p$ is the precision parameter. The covariance matrix is constructed using a squared exponential kernel (Seeger, 2004):

$$K(x, y) = \exp\left(-\frac{(x-y)^2}{2l^2}\right) \tag{18}$$

where $x$ and $y$ are points in the domain, and $l$ is the correlation length, a crucial hyperparameter in our generation process.

**Correlation Length ($l$):** The parameter $l$ controls the smoothness and regularity of the generated PDFs. Specifically:

- Large values of $l$ produce more regular distributions with lower average curvature. These PDFs tend to have smoother features with lower curvatures.
- Small values of $l$ result in more irregular distributions with higher average curvature. These PDFs can have sharper features with higher curvatures.

As $l$ increases, the average curvature decreases, indicating a smoother, less curved function.

**Generation Process:** The random PDF generation involves the following steps:

1. Generate the covariance matrix using the squared exponential kernel.
2. Perform Cholesky decomposition on the covariance matrix.
3. Sample from the Gaussian process using the decomposed matrix.
4. Apply boundary conditions to ensure the PDF goes to zero at the domain edges.
5. Normalize the function to ensure it integrates to 1, making it a valid PDF.

This method allows us to generate a wide range of PDFs with varying degrees of complexity and smoothness, providing a robust test set for our density estimation algorithms. By adjusting the correlation length $l$, we can systematically explore how different estimation methods perform on targets of varying regularity and curvature. Given a randomly generated PDF, we can calculate its average curvature using two methods: numerical differentiation and analytical derivation. Numerically, we can approximate the second derivative using finite differences and then compute the average curvature as:

$$R_{numeric}(p'') \approx \frac{1}{N} \sum_{i=1}^{N} \left(\frac{p(x_{i+1}) - 2p(x_i) + p(x_{i-1})}{(\Delta x)^2}\right)^2 \tag{19}$$

where $N$ is the number of discretization points and $\Delta x$ is the spacing between points. Analytically, we can leverage the fact that the derivative of a Gaussian process is itself a Gaussian process Seeger (2004). For a Gaussian process with squared exponential kernel $k(x, x') = \exp(-\frac{(x-x')^2}{2l^2})$, the expected average curvature can be derived as:

$$R_{analytical}(p'') = \mathbb{E}[\int (p''(x))^2 dx] = \frac{3}{4l^3\sqrt{\pi}} \tag{20}$$

In our numerical experiments, we find that these two methods agree to high precision, typically within 10% relative error, validating our choice to use Gaussian processes to generate PDFs with controllable curvatures.

**Randomly generated distribution at low curvature ($l = 0.5$)**

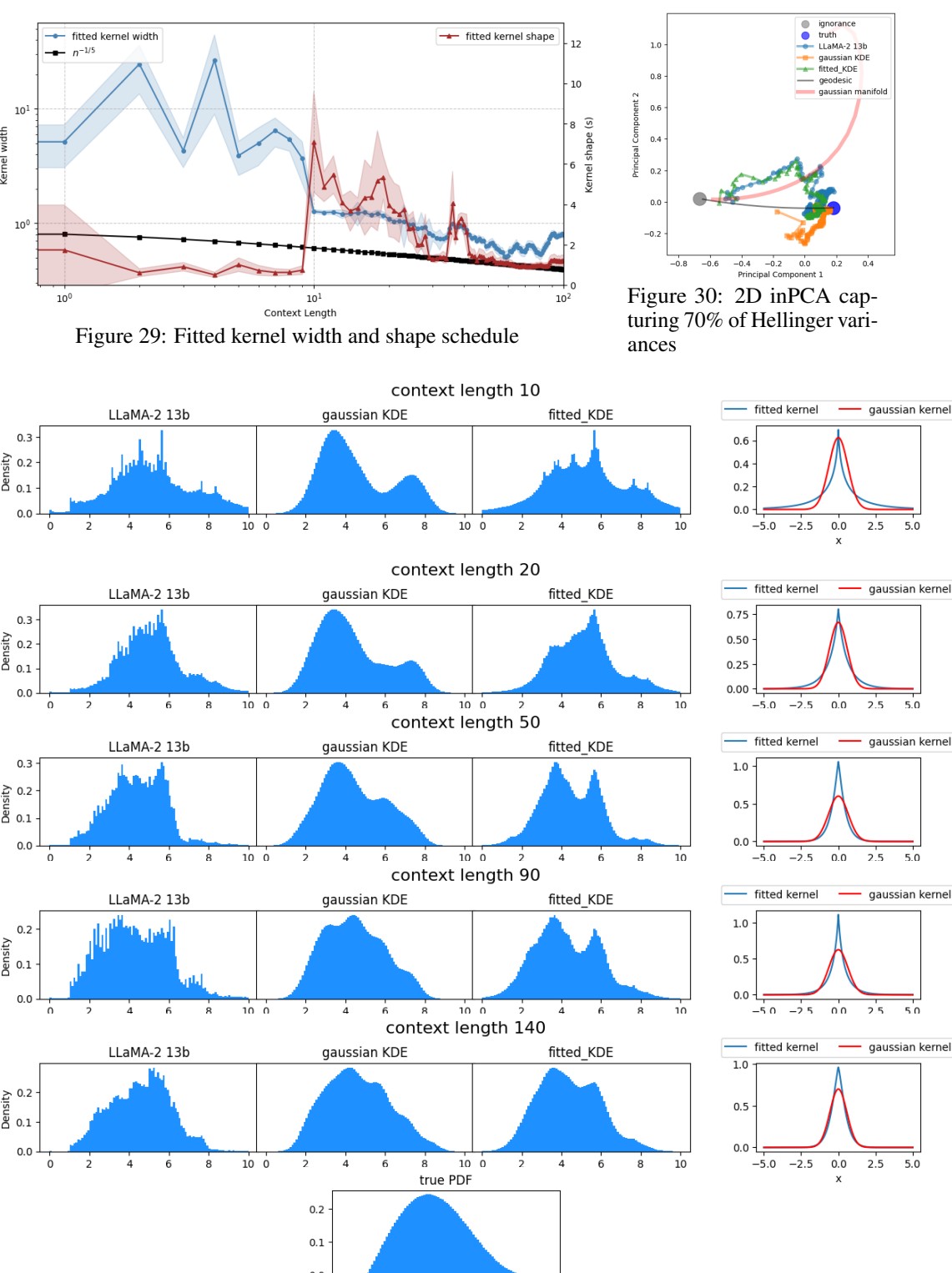

Figure 29: Fitted kernel width and shape schedule

Figure 30: 2D inPCA capturing 70% of Hellinger variances

Figure 31: Fitted kernel visualization with randomly generated target

**Randomly generated distribution at medium curvature** ($l = 0.1$)

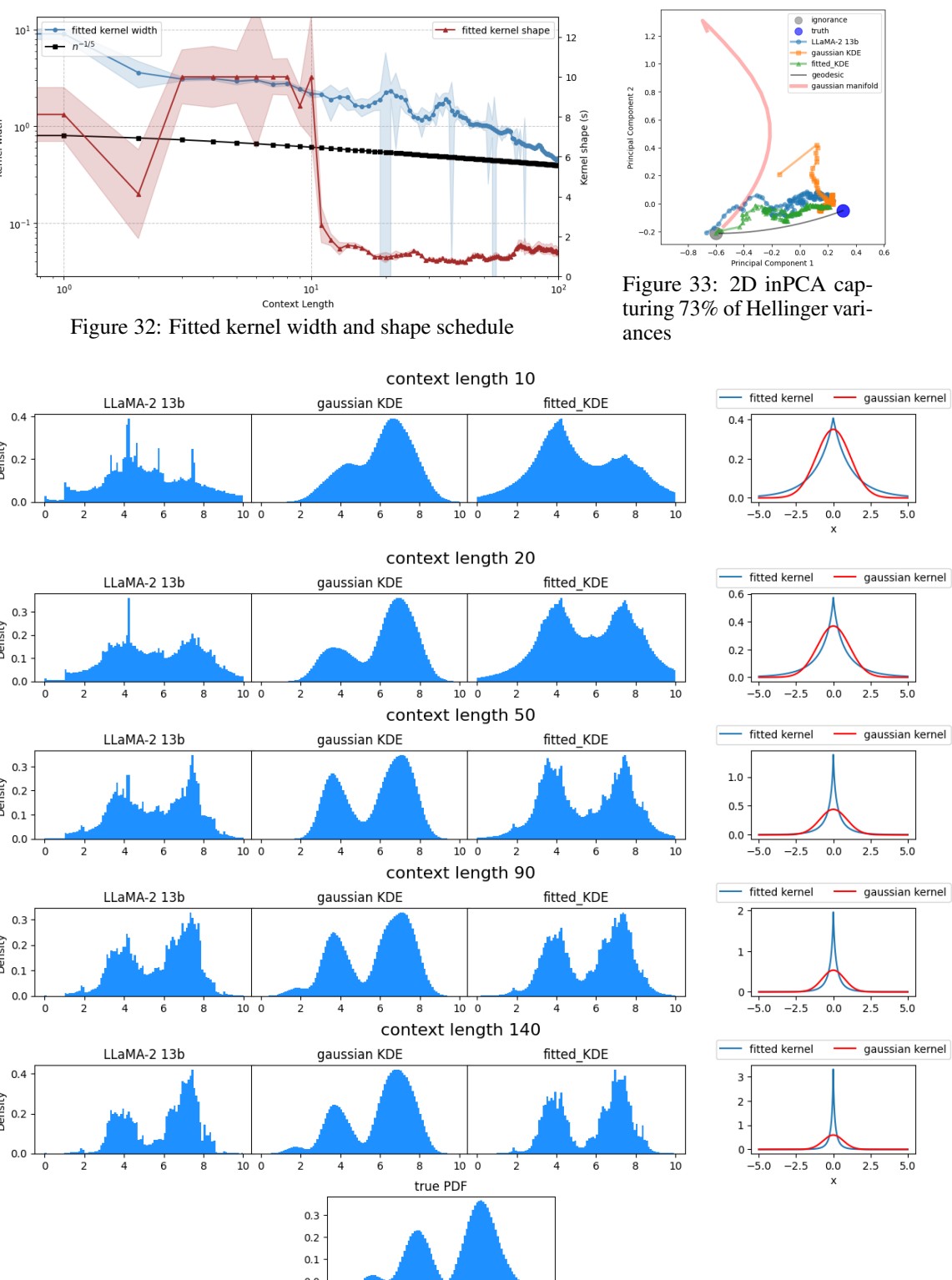

Figure 32: Fitted kernel width and shape schedule

Figure 33: 2D inPCA capturing 73% of Hellinger variances

Figure 34: Fitted kernel visualization with randomly generated target

**Randomly generated distribution at high curvature** ($l = 0.02$)

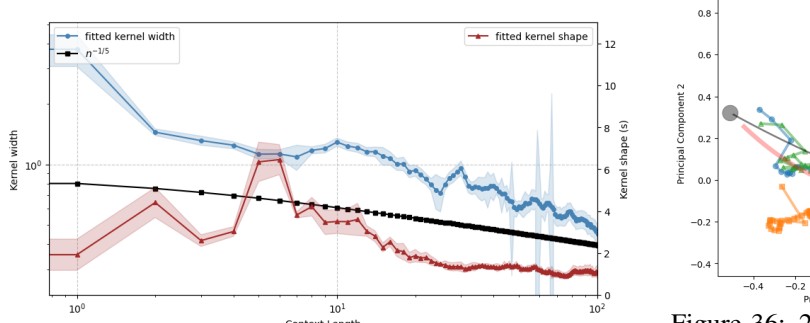

Figure 35: Fitted kernel width and shape schedule

Figure 36: 2D inPCA capturing 55% of Hellinger variances

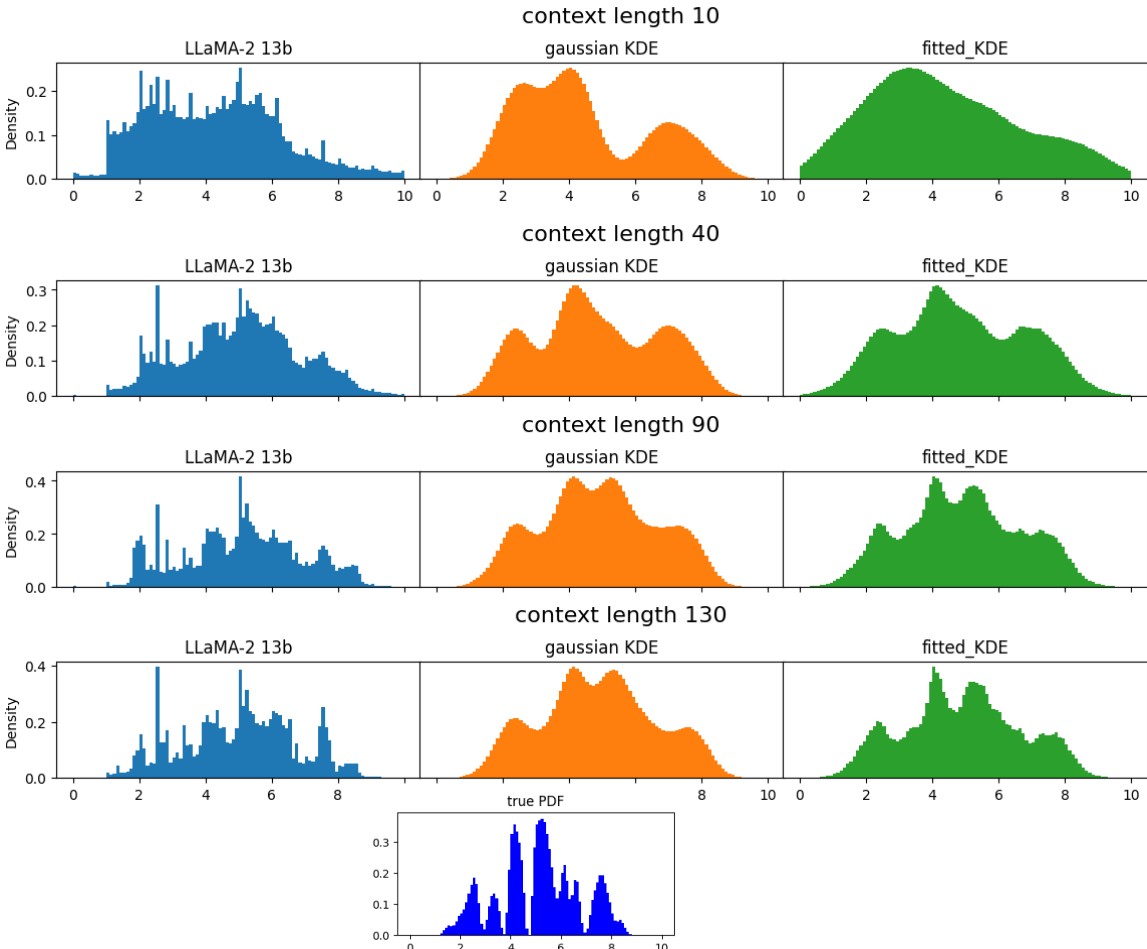

Figure 37: Fitted kernel visualization with randomly generated target

## A.10 HEAVY-TAILED DISTRIBUTIONS

In this section, we visualize the in-context DE trajectories of Student's t-distribution (Ahsanullah et al., 2014). The t-distribution is parametrized as

$$p_v(x) = \frac{\Gamma(\frac{v+1}{2})}{\sqrt{v\pi}\Gamma(\frac{v}{2})} \left(1 + \frac{x^2}{v}\right)^{-\frac{v+1}{2}} \tag{21}$$

where $v$ controls the amount of probability mass in the tails. Smaller $v$ leads to heavier tails. For $v = 1$, the t-distribution reduces to the Cauchy distribution:

$$p(x) = \frac{1}{\pi(1 + x^2)} \tag{22}$$

which has undefined mean and variance due to its extremely heavy tails.

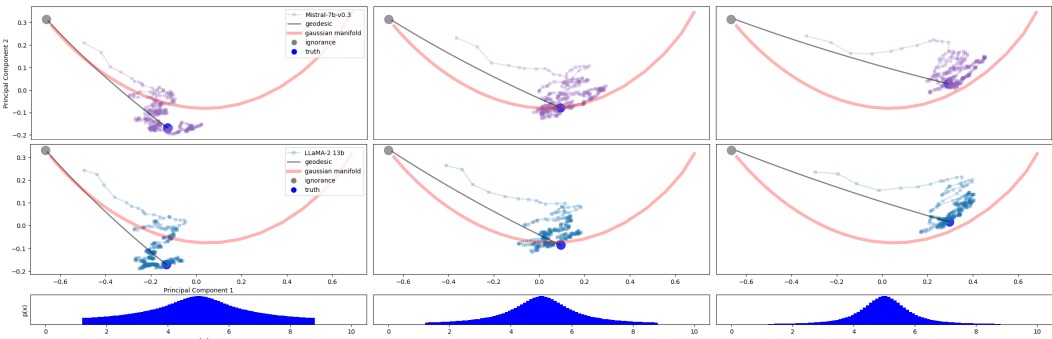

Figure 38: In-context density estimation trajectories for t-distribution targets. Top row: 2D InPCA embeddings of DE trajectories for Mistral-7b-v0.3. Middle row: 2D InPCA embeddings of DE trajectories for LLaMA-2 13b. Bottom row: Corresponding ground truth distributions, with $v = 0.1$, 1, and 2. These 2D embeddings capture 84% of pairwise Hellinger distances between probability distributions.

LLaMA and Mistral trace out similar in-context DE trajectories, as shown in Figure 38. Both models eventually converge to the ground truth distribution. However, unlike the cases with Gaussian and uniform targets, the trajectories for learning t-distributions are more erratic. This is likely due to the fact that t-distribution with small $v$ has large variance (diverging if $v \leq 1$), where the thick tails have non-trivial probability mass. This in turn makes extreme samples occur more frequently, causing instability in any kernel-like density estimators.

## A.11 ADDITIONAL NOTES ON BAYESIAN HISTOGRAM

The Bayesian histogram with uniform prior starts near ignorance by construction, and gradually converges to ground truth distribution with increasing data. Its DE trajectories serve as informative visual comparisons against kernel-like methods. This work does not focus on Bayesian histogram per se. However, for completion, we briefly visualize the learning trajectories of Bayesian histograms at different levels of prior bias, $\alpha$, as defined in Equation 5. This results are shown in Figure 39

Unburdened by a strong prior, Bayesian histograms with smaller $\alpha$ converge faster to the ground truth. However, its learning trajectories are also more erratic. This is because in the absence of a strong regularization, any new data point would cause significant update in the estimated distribution (height of the corresponding bin). This effect is especially pronounced for high-variance distribution, where extreme data occur more often.

Bayesian histograms with larger $\alpha$, on the other hand, follow the geodesic more conservatively in smooth trajectories. However, they also take more data to converge to the true distribution. Note that unlike the other visualizations in this work, each trajectory in Figure 39 consists of $n = 500$ data points, instead of 200.

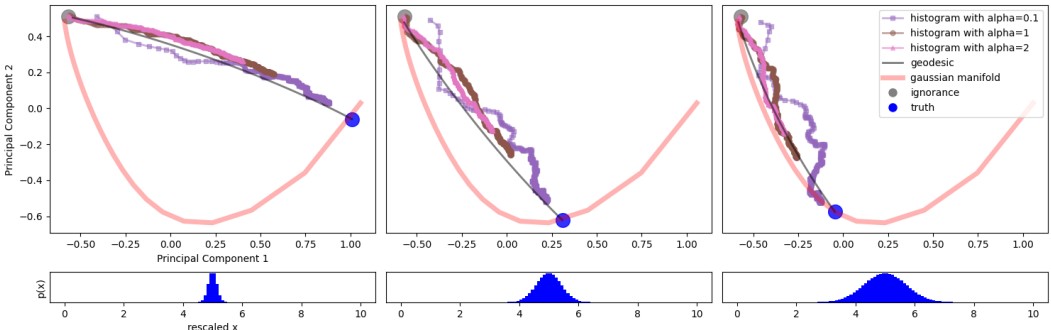

Figure 39: Density estimation trajectories for Bayesian histogram at different uniform bias $\alpha$. Top row: 2D InPCA embeddings of DE trajectories for Mistral-7b-v0.3. Bottom row: Corresponding ground truth distributions. These 2D embeddings capture 87% of pairwise Hellinger distances between probability distributions.

### A.12 UNCERTAINTY QUANTIFICATION FOR BESPOKE KDE PARAMETERS

In this section, we provide information-theoretic justifications for our estimated uncertainty of the fitted KDE parameters, outlined in Section 4.3.1. On a high level, our method is based on the observation that the Hessian matrix at optimal fit is approximately equal to the Fisher information matrix (Kullback, 1997).

Let $P(x)$ denote an LLM's estimated PDF from data, which we treat as the ground truth to fit to. Let $Q_\theta(x)$ denote the PDF estimated by parametrized KDE with parameter vector $\theta = (h, s)$. Technically, both $P(x)$ and $Q_\theta(x)$ are conditioned on the set of data observed in-context, $\{X_i\}_{i=1}^n$. For simplicity, we drop this conditioning in our notation.

During the fitting process, the loss function is defined as the KL-divergence between $P(x)$ and $Q_\theta(x)$:

$$D_{\text{KL}}(P\|Q_\theta) = \int P(x) \log \frac{P(x)}{Q_\theta(x)}\,dx \tag{23}$$

The second derivative (Hessian) of the loss function with respect to $\theta$ is:

$$\nabla_\theta^2 D_{\text{KL}}(P\|Q_\theta) = -\int P(x)\nabla_\theta^2 \ln Q_\theta(x)\,dx$$

The Fisher Information Matrix (FIM) is defined as:

$$I(\theta) = -\mathbb{E}_{Q_\theta}\left[\nabla_\theta^2 \ln Q_\theta(x)\right] = -\int Q_\theta(x)\nabla_\theta^2 \ln Q_\theta(x)\,dx$$

Therefore, at the optimal fit $\theta^*$, if $Q_\theta(x) \approx P(x)$, then the Hessian of the KL divergence provides a good approximation to the FIM:

$$H_{\text{KL}}(\theta^*) = \nabla_\theta^2 D_{\text{KL}}(P\|Q_\theta) \approx I(\theta^*)$$

The covariance matrix of $\theta$ can then be estimated as the inverse of the FIM, a relationship known as the Cramér-Rao bound (Thomas & Joy, 2006). Under regularity conditions and asymptotic normality, this bound becomes an equality:

$$\text{Cov}(\theta) = I(\theta^*) \approx \left(\nabla_\theta^2 D_{\text{KL}}(P\|Q_\theta)\right)^{-1}$$

Said another way, the uncertainty in the fitted kernel parameters, such as bandwidth (h) and shape (s), can be estimated from the inverted Hessian matrix of KL-divergence loss function.

### A.13    KDE WITH SILVERMAN BANDWIDTH SCHEDULE

For the majority of our investigations, we have been comparing LLaMA's in-context DE process with Gaussian KDE with bandwidth schedule

$$h(n) = Cn^{-\frac{1}{5}}$$

with $C = 1$. However, as noted in Appendix A.7.1, in practice one often estimates the pre-coefficient from data, such as Silverman's rule of thumb (Equation 17). In this section, we replicate Figures 3 and 4 from Section 4, and use Silverman's rule-of-thumb bandwidth instead of $C = 1$. As shown in Figures 40 and 41, even with Silverman's pre-coefficient, Gaussian KDE still shows a clear bias towards the Gaussian submanifold.

**Gaussian target**

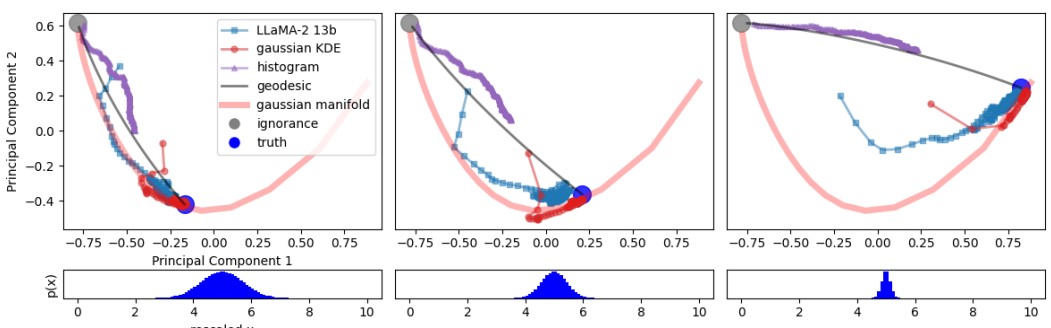

Figure 40: In-context density estimation trajectories for Gaussian targets. Top row: 2D InPCA embeddings of DE trajectories for Gaussian targets of decreasing width (left to right). Bottom row: Corresponding ground truth distributions. These 2D embeddings capture 94% of pairwise Hellinger distances between probability distributions.

**Uniform target**

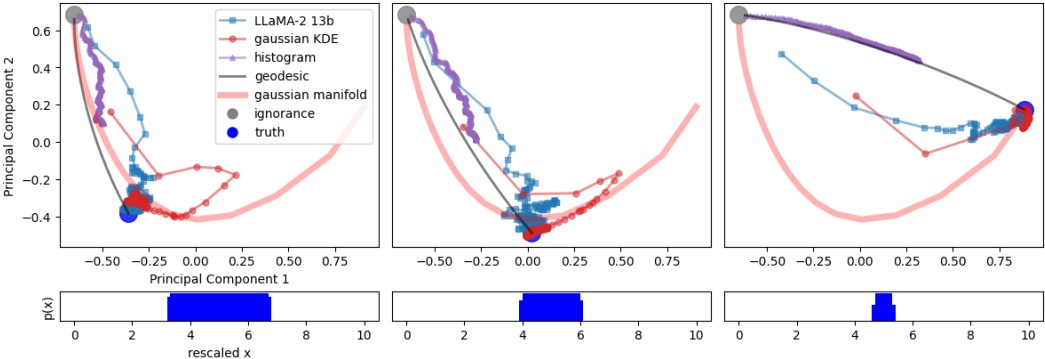

Figure 41: In-context density estimation trajectories for Gaussian targets. Top row: 2D InPCA embeddings of DE trajectories for Gaussian targets of decreasing width (left to right). Bottom row: Corresponding ground truth distributions. These 2D embeddings capture 91% of pairwise Hellinger distances between probability distributions.

With Silverman's heuristic bandwidth, the Gaussian KDE algorithm can now efficiently converge to the ground truth target distribution, performing on par with or even surpassing LLaMA-13b's in-context DE in terms of speed. This outcome is not surprising for two reasons: 1. LLaMA is a general-purpose LLM that is not optimized for density estimation tasks. 2. We do not specifically prompt LLaMA to perform DE, so it must infer the task from the raw data sequence. This comparison is inherently unfair to LLaMA, as Gaussian KDE is specifically designed for DE tasks.

We would like to reiterate that the purpose of these visualizations is not to benchmark LLaMA against existing algorithms, but to distill geometric insights. The Gaussian KDE, with Silverman's bandwidth, features much shorter trajectory lengths, and therefore provides fewer geometric insights compared to our previous visualizations with fixed bandwidth.

