# OpenReview forum: "Density estimation with LLMs: a geometric investigation of in-context learning trajectories"
_ICLR.cc/2025/Conference — ICLR 2025 Poster_

### Official Review · Reviewer_2CMq · 2024-10-20

**Soundness:** 3
**Presentation:** 2
**Contribution:** 3
**Rating:** 6
**Confidence:** 4

**Summary:**

The paper titled “Density Estimation with LLMs: A Geometric Investigation of In-Context Learning Trajectories” investigates the ability of large language models (LLMs), specifically LLaMA-2 models, to perform density estimation (DE) from in-context data. Density estimation is crucial for probabilistic modeling, where the goal is to estimate a probability density function (PDF) based on observed data. The authors leverage Intensive Principal Component Analysis (InPCA) to visualize and analyze the in-context learning dynamics of these models, showing that LLaMA-2 follows distinct low-dimensional trajectories in probability space when compared to traditional methods such as histograms and Gaussian kernel density estimation (KDE). A key insight from the paper is that LLaMA-2’s in-context DE process can be interpreted as an adaptive KDE with a dynamic kernel shape and width, which captures the model’s behavior with high precision. This adaptive KDE model, with only two parameters, successfully explains the geometric features of the model’s DE trajectories, offering a deeper understanding of how LLMs perform probabilistic reasoning in-context. The paper also speculates that LLaMA-2 may use a mechanism similar to the dispersive induction head to adjust predictions based on observed data.

**Strengths:**

Originality: The paper contributes a novel approach to understanding the in-context learning capabilities of large language models (LLMs) by framing their density estimation (DE) processes as an adaptive form of kernel density estimation (KDE). This is an innovative application of KDE in a context that diverges from traditional statistical methods. The use of Intensive Principal Component Analysis (InPCA) to visualize and interpret learning trajectories in probability space adds further originality, combining ideas from both geometric analysis and machine learning in a fresh way. The concept of interpreting LLaMA’s learning trajectory as a kernel-like algorithm, with an adaptive kernel, is particularly creative and represents an advancement in understanding how LLMs process probabilistic information.

Quality: The research is rigorous and well-supported by both theoretical and empirical analyses. The application of InPCA, along with detailed comparisons between LLaMA’s trajectories and traditional DE methods like histograms and Gaussian KDE, demonstrates the authors’ thoroughness in exploring the model’s behavior. The paper is grounded in solid mathematical foundations, using well-established statistical measures such as the Hellinger distance to quantify and compare learning trajectories. The methodology is well-executed, and the visualizations (e.g., 2D embeddings of DE trajectories) are informative and enhance the quality of the research.

Clarity: The paper is generally clear in its explanations of complex concepts, especially given the technical nature of the topic. The authors effectively explain the connection between KDE and LLMs’ in-context learning, providing sufficient background for the reader to follow the argument. The figures and visualizations, such as those illustrating LLaMA’s DE process and its comparison with classical methods, are well-designed and help clarify the main findings. However, some sections, particularly around the technical details of InPCA and bespoke KDE modeling, may require more careful reading by less specialized audiences.

Significance: The significance of the work lies in its contribution to the growing field of in-context learning for LLMs. By providing a new geometric and probabilistic interpretation of LLaMA’s behavior, the paper has the potential to impact future research on both LLMs and kernel-based methods. The idea of modeling LLMs’ probabilistic reasoning with an adaptive kernel framework offers a new lens through which to study these models, which could inform the development of more efficient or interpretable LLMs in various domains. Moreover, the paper opens avenues for future exploration, such as investigating the role of dispersive induction heads in continuous probabilistic learning, making it a significant contribution to both theoretical and practical applications in AI.

**Weaknesses:**

While the paper “Density Estimation with LLMs: A Geometric Investigation of In-Context Learning Trajectories” offers a strong contribution, there are several areas where it could be improved.

1. Insufficient exploration of broader datasets: The paper focuses primarily on Gaussian and uniform distributions for testing LLaMA’s in-context density estimation (DE) capabilities. While these distributions are useful for illustrating the model’s ability to perform DE, they are relatively simple and may not fully capture the complexity of real-world probabilistic learning tasks. Incorporating more complex and diverse target distributions, such as multimodal or skewed distributions, would significantly strengthen the generalizability of the findings. By broadening the scope of the experiments, the authors could better demonstrate the robustness of LLaMA’s DE across a wider variety of data structures.

2. Incomplete explanation of adaptive kernel choices: The core contribution of the paper is the interpretation of LLaMA’s in-context DE process as an adaptive KDE, but the reasoning behind the specific choices of kernel shapes and widths, and their relationship to LLaMA’s behavior, could be made clearer. For instance, while the authors introduce a flexible kernel with a shape parameter, the empirical results seem to focus primarily on the Gaussian kernel and only touch briefly on other possibilities. Expanding this section to include more detailed justifications for the selection of kernel parameters and providing more comparative results with different kernel shapes would offer deeper insights into why LLaMA behaves as it does.

3. Theoretical depth versus empirical validation: The paper proposes several intriguing theoretical concepts, such as the idea of dispersive induction heads, but these remain speculative without much empirical validation. The authors propose that LLaMA employs a dispersive induction mechanism to handle in-context density estimation, yet this idea is not thoroughly tested or backed up with concrete evidence. To improve the work, the authors could design specific experiments or ablation studies to directly probe the existence of this mechanism, or alternatively, provide more substantial theoretical evidence. This would move the idea from speculation to a more solid theoretical contribution.

4. Comparative analysis with more models and baselines: The paper primarily compares LLaMA’s performance against classical KDE and Bayesian histograms. However, it would benefit from including a broader set of baseline models, including other transformer-based LLMs or models specifically designed for probabilistic reasoning. For instance, comparing LLaMA’s performance against other architectures known for in-context learning abilities could provide more insight into how LLaMA’s density estimation compares to other state-of-the-art models. This would make the experimental results more compelling and allow for a better understanding of LLaMA’s unique contributions.

**Questions:**

1. Clarification on Kernel Selection and Adaptation: How were the kernel shapes and bandwidth schedules chosen, and why were they deemed the most appropriate for LLaMA’s in-context learning trajectories? Did you experiment with non-Gaussian kernel shapes beyond those briefly mentioned?

2. Scope of Data Distributions: Why did you primarily focus on Gaussian and uniform target distributions? How might LLaMA’s in-context DE capabilities generalize to more complex, real-world distributions, such as multimodal or skewed distributions?

3. Empirical Validation of Theoretical Concepts:How can you empirically validate the proposed concept of “dispersive induction heads”? What experiments would directly test this idea beyond the adaptive KDE model?

4. Comparison with Other Transformer Models: How does LLaMA-2’s performance in in-context DE compare to other large language models  or other architectures known for probabilistic reasoning?

5. Generalization Beyond DE Tasks: How do you see the techniques and findings in this paper extending to tasks beyond density estimation, such as other forms of probabilistic reasoning or inference? Could these methods be generalized to conditional density estimation or more complex probabilistic models?

6. Impact of Tokenization on DE Process: How does the tokenization process impact the DE process in LLaMA? Could different tokenization schemes affect the observed DE trajectories or results?

7. Role of Model Size: How does the model size (7B, 13B, 70B) impact the in-context density estimation process? Are there any trends or differences in the DE trajectories of these models, and do larger models perform better in this context?

---

> ### Author Response · Authors · 2024-11-17
>
> We are grateful to the reviewer for the very detailed and insightful review. We are pleased to note that the reviewer appreciated the “novel approach”, rigor of the research “well-supported by both theoretical and empirical analyses”, clarity of the paper, and that this work is “a significant contribution to both theoretical and practical applications in AI”. We provide a response to the reviewer’s comments and questions below.
>
> **Broader datasets**
>
> We thank the reviewer for suggesting additional experiments to clarify this point.
>
> Currently, the main body of the paper only discusses experiments with simple symmetric distributions because these are, so far, the only cases where the learning trajectory admits faithful 2D embeddings, where the distances between the embedded points accurately reflect the statistical distance between the corresponding distributions.. In appendix A.7, we include experiments on randomly-generated ground-truth distributions, some of which are multi-modal, for example, the one on page 26. The one on page 25 is approximately a skewed distribution. The LLM is indeed able to estimate all of these distributions, with the DE trajectory well-approximated by the adaptive KDE model. It’s just that the 2D trajectory visualization provides little insights on these more complex cases.
>
> In the updated appendix, we will include additional experiments on explicitly bimodal distributions, such as as Gaussian mixture suggested by reviewer acu3, as well as heavy-tailed distributions, such as Cauchy, as suggested by reviewer mikv. We will also include skewed distributions suggested by the reviewer.
>
> **Choice of Gaussian kernel for comparison**
>
> We thank the reviewer for the comment.
> Among our main findings is the fact that LLM’s in-context DE process could be interpreted as an adaptive KDE. This finding is independent of the Gaussian KDE, which only serves as a visual comparison in the InPCA analysis. As explained in the paragraph 331 to 336, many kernel-based density estimators share a similar bias towards Gaussianiety, as further illustrated in figure 13 of the Appendix. In the main paper, therefore, we use Gaussian KDE as a stand-in for all conventional kernel methods.
>
> In our updated manuscript, we will clarify the role played by the Gaussian KDE, to avoid potential miscommunication regarding our main findings.
>
> **How to verify dispersive induction heads**
>
> Our paper focuses on showing how real foundation models perform probabilistic reasoning, and that a kernel-like algorithm emerges from simple next-token prediction training. Eliciting its mechanistic basis is therefore beyond the scope of our current work.
>
> That being said, we thank the reviewer for raising a highly interesting and non-trivial question for future research. In the mechanistic interpretation community, researchers typically train smaller and more interpretable models, and then use linear probing and ablations to isolate the circuit of interest.
>
> Note that the mechanistic basis of the original induction head was quite complicated, and took the course of several separate works to develop (explained in Appendix A.1.). We therefore think it might be appropriate to focus on introducing the dispersive induction head concept in our current paper, and leave the door open for researchers to better understand the mechanistic basis of it.
>
> **Comparison with Other Transformer Models**
>
> We agree with the reviewer’s suggestion that experiments on other models could help make our results more comprehensive. We are currently working on repeating our experiments on Gemma and Mistral, and will share the results as soon as we can. However, we would like to emphasize that our work is not to be interpreted as a benchmarking paper. Our goal is to understand how LLMs perform data-driven probabilistic modeling, instead of which LLMs is the most performant, or how they compare to other established methods.
>
> **From density estimation to probabilistic reasoning**
>
> We regard density estimation as a cornerstone problem that underpins many data-driven probabilistic modeling tasks such as time-series prediction, and conditioned densities estimation. These connections are discussed in section two: background, specifically the paragraph from 101 to 110.
>
> A recent line of work investigates the possibility of using LLMs to make probabilistic predictions conditioned on language [1]. For example, one could provide the LLMs with natural language hints before feeding the serialized numerical data, for example “the following is a series of numbers sampled independently from a distribution: …”
>
> [1]: Requeima, J., Bronskill, J.F., Choi, D., Turner, R.E., & Duvenaud, D.K. (2024). LLM Processes: Numerical Predictive Distributions Conditioned on Natural Language. ArXiv, abs/2405.12856.
>
> A natural future direction, therefore, is to study how the adaptive kernel changes based on natural language instructions.

---

> > ### Author Response · Authors · 2024-11-17
> >
> > **Impact of Tokenization on DE Process**
> >
> > Yes, tokenization schemes do impact the in-context DE process, and specifically, how one should extract the probabilistic prediction from an LLM. Gemma, Mistral, and LLaMA-2 are compatible with the Hierarchy-PDF method leveraged in this paper, because their tokenizers uniquely express multi-digit numbers. For example, the number “123” is tokenized as three independent tokens representing “1”, “2”, and “3”. This is not the case for other models, such as LLaMA-3, which could also express this same number as the token for “12” followed by “3”, or the token for “1” followed by “23”. Such degenerate cases make the application of the Hierarchy PDF method more involved.
> >
> > **Role of Model Size**
> >
> > This is an important point that deserves a note in the main body of the updated manuscript. We thank the reviewer for pointing this out.
> >
> > While the main results focus on 13B, in Figure 2 part C, we visualize the in-context DE process of all three models from the LLaMA suite. We also visualize the trajectory-wise distance between these models in Figure 11 of the Appendix, via what we call the meta-inPCA embedding. It could be observed that, while there are slight variabilities between 7B, 13B, 70B, these variabilities are miniscule compared to the difference between in-context DE and classical methods such as Gaussian KDE and Bayesian histogram. Furthermore, the adaptive KDE model is also able to explain the variabilities between the three LLaMA models.

---

> > > ### Author Response · Authors · 2024-11-23
> > > **Thank you, and looking forward to the reviewer's response**
> > >
> > > We thank the reviewer for the comments and suggestions, which have helped up to substantially strengthen our paper, as well as expand its scope. We hope that our added experiments, visualizations, and explanations have adequately addressed your questions and concerns.
> > >
> > > As the author-reviewer discussion period is coming to an end, we would be happy to read their reply to answer any additional questions. After that, we may not have the opportunity to respond to your comments.
> > >
> > > If you believe we have addressed your comments, we would be grateful if you could acknowledge this and update your score. Otherwise, we are happy to provide further clarification.
> > >
> > > Thank you again for your time and valuable review.

---

### Official Review · Reviewer_acu3 · 2024-11-02

**Soundness:** 3
**Presentation:** 3
**Contribution:** 3
**Rating:** 8
**Confidence:** 4

**Summary:**

This work explores how LLMs infer probability density functions, derived from in-context observations. It uses a variant of Principal Component Analysis to help visualise the learning dynamics. Llama-2 models are chosen for the experiments, and these are fed data from target distributions which are either uniform or Gaussian. The results are found to be comparable to certain forms of kernel density estimation.

**Strengths:**

The paper is well written and clearly presented.

The approach is novel, and the topic is of high interest.

There is a large number of supportive experiments and these are presented with well chosen visualisations.

**Weaknesses:**

I have a few concerns regarding the scope of the experiments:

- Models
The empirical results would be significantly more compelling if they weren't all using the Llama-2 family of models, as then the results could potentially be claimed to apply more generally to LLMs. As it stands, the title of the paper may be considered slightly misleading since conclusions can only be drawn on this specific family of models, and not LLMs in general.

- Distances
When visualising the results, it would be beneficial not to solely rely upon the Hellinger distance but to also show results derived from other distances such as the KL divergence or Wasserstein distance. Even if this does not lead to such compelling visualisations, this would provide a much more complete picture of how well the predicted distributions align with the targets.

- Distributions
The chosen target probability distributions are very simple (Gaussian & uniform) so this also limits the conclusions the reader can draw. How would the LLM respond to data from a bimodal distribution, for example?

- Uncertainties
In order to establish a robust, reproducible set of empirical results, it would be strongly advisable to assign uncertainty estimates to the key quantities involved, such as the bespoke KDE parameters.

**Questions:**

While my main concerns relate to the empirical results, as outlined in the 'weaknesses' section, I have some smaller questions below:

> a uniform distribution over the domain (0, 100), showing a state of neutral ignorance, which is a reasonable Bayesian prior
I'm not so sure that is a reasonable Bayesian prior. To quote David Mackay's book, "ignorance does not correspond to a uniform probability distribution". I would be surprised if Llama has a priori a uniform prior over the digits, since 1 is much more often the leading digit than others.
As David MacKay argues in his book on Information Theory, "ignorance does not correspond to a uniform probability distribution". Given that lower digits appear more frequently as leading digits (in line with Benford's Law), it would be surprising if LLMs such as LLaMA  adopted a uniform prior over digits without explicit conditioning. Could this be the reason we can see in Figures 2 and 3 that the Llama trajectory starts some distance from the grey point marked as 'ignorance'?

> While LLaMA-2 has a context window of 4096 tokens (equivalent to ∼1365 comma-delimited, 2-digit data points), we limit our
analysis to a context length of n = 200.
Could you clarify here whether n=200 relates to max number of tokens or data points? (I would interpret this sentence to imply n = 200 numbers, except I notice Figure 1 seems to illustrate 200 datapoints which would correspond to many more than 200 tokens)

The paper frequently refers to the target function as a probability density function (PDF). However, since the outputs are discrete values, perhaps it would be more accurate to describe this function as a probability mass function (PMF)?

---

> ### Author Response · Authors · 2024-11-17
>
> We thank the reviewer for the detailed feedback and noting that the “paper is well written” and that the “approach is novel and the topic is of high interest”. We provide below point-by-point answers to the different comments and suggestions, detailing the additional experiments we chose to perform following the reviewer’s suggestions. Since the experiments are computationally intensive, we will be including them in the paper throughout the discussion period as they come.
>
> **Models beyond LLaMA-2**
>
> This is an excellent suggestion which will indeed make the paper stronger. Following the reviewer’s comment, we are currently working on repeating our experiments on Gemma and Mistral models (released this year), and will share the results as soon as we can.
>
> The choice of these additional models was made because they share the same tokenizer as Llama-2 (unlike Llama-3), where each multi-digit number is represented uniquely as a succession of 3 tokens (e.g., the number “123” is tokenized as three independent tokens representing “1”, “2”, and “3”). This is not the case for Llama-3 models, which makes the application of the algorithm Hierarchy PDF for extracting the PDF more involved. We will add a comment about this in the revised manuscript.
>
> **Choice of distances for InPCA**
>
> This is a great comment, Appendix 5.3 provides some explanations regarding the choice of the Hellinger distance (which gave us the most interpretable geometric interpretation in the early experiments that led to this work).
>
> We note that the typical KL divergence is not applicable to inPCA visualization, since it is not symmetric, and hence not a true mathematical distance: the KL divergence from P to Q is not the same as Q to P, rendering a geometric embedding of P and Q impossible. That being said, we do agree that adding some comparison with the KL divergence would be interesting to get a more complete picture of the distributions. Moreover, given the popularity of KL divergence in modern machine learning, it makes sense to use that as the distance measure. To that end, we will repeat some of the visualizations using the symmetrized KL-divergence: $D_{SKL}(P,Q) = D_{KL}(P,Q) + D_{KL}(Q,P)$.
>
> **More complex target distributions**
>
> Currently, the main body of the paper only discusses these simple cases because these are, so far, the only cases where the learning trajectory admits faithful 2D embeddings, where the distances between the embedded points accurately reflect the statistical distance between the corresponding distributions. In appendix A.7, we include experiments on randomly-generated ground-truth distributions, some of which are multi-modal, for example, the one on page 26. One can appreciate both from the visualization (Figure 30) and the actual distributions (Figure 31), that the estimated density by LLaMA-2 13b is much similar to the bespoke KDE kernel, and not as similar to the classical Gaussian KDE.
>
> In an updated version of the manuscript, we will include additional experiments on explicitly bimodal distributions, such as as Gaussian mixture suggested by reviewer acu3, as well as heavy-tailed distributions, such as Cauchy, as suggested by reviewer mikv.
>
> **Uncertainty in fitted KDE parameters**
>
> We thank the reviewer for this suggestion. In Figures 6 and all other figures in the manuscript where the adaptive kernel parameters are plotted, we actually visualize the 1-sigma uncertainty of the fitted parameter using shaded regions around the curve. As explained briefly in the paragraph from line 430 to 431, the uncertainty of these fitted parameters are estimated using the inverted Hessian matrix at the point of best fit, which is consistent with the inverse problem community.
>
> We will update the caption of the figure to clarify this point. In our updated appendix, we will also add more theoretical justification for this choice of uncertainty estimation, along the following lines: our uncertainty estimation for the fitted kernel parameters is well-motivated from an information-theoretical perspective. Specifically, when the loss function is the Hellinger distance, or the KL divergence, the Hessian could be interpreted as approximately the fisher information matrix (FIM). The inverse of FIM is a widely accepted estimation of covariance matrix of the fitted parameters.
>
> **Skewed ignorance**
>
> We thank the reviewer for raising this intriguing hypothesis. Staring at Figure 2 part b, it does seem that the left tail is consistently higher than the right. In our updated appendix, we will visualize the models’ “true” prior at the 0 data limit, by probing the model with only the “start of the sentence” token and nothing else.
>
> Another source of non-uniform bias, in our opinion, is that LLMs are slightly biased towards nice integers such as 0, 1, 2, … ,9, which are the small spikes that can be observed in Figure 2 part b.

---

> > ### Author Response · Authors · 2024-11-17
> >
> > **Definition of context length (n)**
> >
> > We thank the reviewer for shedding light on the ambiguity in our terminologies. In our paper, context length n consistently refers to the number of data points. Each data point consists of three tokens (2 digits and 1 comma) . For clarity, perhaps we should change the phrase “context length (n)” to “number of in-context data (n)”. Would this be a better phrasing?
> >
> > **Terminology regarding PDF or PMF**
> >
> > While it is true that the output from LLM are discrete values, the resulting histogram should be interpreted as a PDF with each bin representing a continuous uniform density. It is the total density that summarizes to 1. To put in another way, the y-axis of the hierarchy-PDF object does not represent the probability mass of each bin, but its density, which is probability mass divided by bin width.
> >
> > This interpretation is also consistent with a prior work that pioneered the usage of LLMs for time-series forecasting:
> > Gruver, Nate, et al. "Large language models are zero-shot time series forecasters." Advances in Neural Information Processing Systems 36 (2024).

---

> ### Author Response · Authors · 2024-11-23
> **Thank you, and looking forward to the reviewer's response**
>
> We thank the Reviewer for the comments and suggestions, which have helped up to substantially strengthen our paper, as well as expand its scope. We hope that our added experiments and visualizations have adequately addressed your questions and concerns.
>
> As the author-reviewer discussion period is coming to an end, we would be happy to read their reply to answer any additional questions. After that, we may not have the opportunity to respond to your comments.
>
> If you believe we have addressed your comments, we would be grateful if you could acknowledge this and update your score. Otherwise, we are happy to provide further clarification.
>
> Thank you again for your time and valuable review.

---

> > ### Comment · Reviewer_acu3 · 2024-11-23
> >
> > The revised submission certainly marks a major improvement over the original, with the new LLMs, the more complex pdfs including bimodality, and the additional distance metrics.
> >
> > > change the phrase “context length (n)” to “number of in-context data (n)”. Would this be a better phrasing?
> >
> > Yes I agree, that would be a clearer phrasing in my view.
> >
> > > slightly biased towards nice integers
> >
> > I think that's a useful observation, it seems reasonable that the LLM's prior corresponds to a superposition of a continuous part and those spikes, to represent integers and real numbers.
> >
> > > In our updated appendix, we will visualize the models’ “true” prior at the 0 data limit, by probing the model with only the “start of the sentence” token and nothing else.
> >
> > Yes that is a very helpful addition, which I see is now in Figure 25. Do you know what is causing the LLM pdf to slightly peak at '5' even before any data is seen, is this just a coincidence?

---

> ### Author Response · Authors · 2024-11-24
>
> Our apologies, due to a glitch in our plotting pipeline, the first plot in Figure 25 was actually at (n=5).
> It has been corrected now. As shown in the updated Figure 25, the LLM's prior at (n=0) is indeed a uniform distribution with peaks at "nice" integers.
>
> Thanks for catching this!
>
> > It seems reasonable that the LLM's prior corresponds to a superposition of a continuous part and those spikes, to represent integers and real numbers.
>
> Thanks again for the insightful discussion. While we are mostly changing the appendices during the reviewer-author discussion period, if accepted, we will update the manuscript to highlight this observation, as well as the lopsidedness towards lower numbers, potentially due to Benford's Law.
>
> We are glad that you found our update manuscript significantly improved. We'd be grateful if you would consider increasing your score if our additional experiments and explanations have addressed your questions.

---

> > ### Comment · Reviewer_acu3 · 2024-11-26
> >
> > Thank you for the update, it's good to see n=0 is fixed!  And yes I will update my score accordingly.
> >
> > One other small suggestion - it might be interesting to mention the impact of temperature. Since the paper uses the raw softmax predictions, it implicitly assumes T=1, but it might be interesting to discuss or show what happens as we reduce T, thereby increasing the sharpness of the pdfs.

---

> > > ### Author Response · Authors · 2024-11-26
> > >
> > > Thank you for taking the time to consider our responses. We are glad we were able to address your comments adequately.
> > >
> > > > The impact of temperature
> > >
> > > This is a great suggestion. We have not tried temperature scaling since we have been primarily focusing on the setup that the models were trained on, in order to avoid manually tuning the output PDFs of the model. We will do an experiment to see the effect of the temperature for the appendix of the camera-ready version.

---

> > > > ### Comment · Reviewer_acu3 · 2024-11-26
> > > >
> > > > I'm glad you found the suggestion to be useful - it would be particularly intriguing if the T < 1 values commonly used for text generation are found to offer objectively better statistical performance. I see the discussion period has been extended, so if you happen to obtain any preliminary results by then, do post them here.

---

### Official Review · Reviewer_KbiX · 2024-11-03

**Soundness:** 3
**Presentation:** 2
**Contribution:** 3
**Rating:** 6
**Confidence:** 3

**Summary:**

This paper investigates the behavioral characteristics of LLMs (more specifically, LLaMA-2) in performing probability density estimation from in-context by using InPCA. It finds that compared to traditional KDE and Bayesian histogram estimation, the proposed KDE with adaptive kernel width and shape resembles their behavior more closely.

**Strengths:**

1.	Probability density estimation is a classic and important problem in statistics and this paper explores the underlying mechanism of utilizing LLMs' in-context capabilities for probability density estimation, which is novel to me.

2.	The authors designed various experiments that are easy to understand, providing support for their viewpoints and interesting findings regarding the potential patterns of density estimation in LLMs.

3.	The paper has a clear structure and is easy to follow.

**Weaknesses:**

1.	Although using LLMs for probability density estimation is relatively novel compared to traditional methods, the scope of the discussion in the paper is somewhat limited (for example, it focuses on the one-dimensional case where continuous probability densities are discretized into 2-digit numbers). It would be interesting to explore whether LLMs can still make effective predictions and the analysis results remain consistent in more complex scenarios.

2.	When estimating more complex distributions, such as those with higher curvature (l = 0.02) as shown in Figures 32 - 34, the context length may limit the model's estimation performance. It would be worth investigating whether increasing the context length would lead to better estimation results.

**Questions:**

1.	It is observed in part b of Figure 2 that, for both n = 10 and n = 50, the estimated density functions appear to have an upward bias at the edges (when x is close to 0 and 10). Could the authors provide some explanation for this?

By the way, it can be seen that when the context length is limited in Figure 2 b, the LLMs' estimation of the Gaussian distribution seems to resemble the scenario where the proposed bespoke KDE method has a smaller s (depicted in Figure 5, s = 1 for example). Therefore, as seen in Figure 13, the estimation using the exponential KDE appears to be closer to LLaMA's prediction pattern than that using the Gaussian KDE. This might also provide us with some insights.

2.	When using Bayesian histogram to estimate a Gaussian distribution, the authors propose that its slow convergence is 'likely due to the strong influence of its uniform prior'. However, when estimating a uniform distribution, it does not seem to converge faster either. Therefore, this explanation is somewhat contradictory. Furthermore, what would happen if \alpha is not set to 1 (for example, if \alpha = 0)?

Typos:
It seems like there is a missing citation (squared exponential kernel (?)) in line 1253.

---

> ### Author Response · Authors · 2024-11-17
>
> We thank the reviewer for the insightful review and the positive feedback of the novel approach of using LLMs for PDF estimation explored in this work. We provide below a detailed point-by-point answer to the reviewer’s comments.
>
> **More complex settings**
>
> While we agree with the reviewer that the experiments focus on relatively simple tasks, this is mainly because we are interested in understanding “how” LLMs grapple with probabilistic reasoning tasks, and these mechanistic interpretations are better elucidated with distributions that admit intuitive visualizations.
>
> As also remarked by reviewer mikv, the idea to generalize the observation to the multivariate case is certainly interesting. However, as we explained in the response to the comment, generalizing the PDF estimation algorithm to multivariate cases has technical challenges since LLMs are not “permutation invariant” with respect to the ordering of multivariate variables. In addition, we are currently repeating some of the experiments using 3-digit representations, to investigate whether the enhanced precision leads to consistent results.
>
> **Estimating high curvature PDF with longer context length**
>
> We thank the reviewer for pointing out the possibility of non-convergence when learning complex distributions. We are currently repeating the experiments of high curvature random distribution using maximum context length (~1000 states). In our updated appendix, we will plot the loss function (Hellinger distance or KL divergence) as a function of context length. In our earlier experiments, these loss curves typically plateau after at around 400 states.
>
> **Bias towards the edges**
>
> This is indeed an interesting artifact that deserves a note in the paper, and we appreciate the reviewer for pointing it out.
>
> It turns out that in Figure 2 part b, there are spikes in the predicated PDF not only at 0.0 and 10.0, but also at other “nice” numbers such as 1.0, 2.0, … 9.0. A possible cause for this, in our opinion, is that these rounded numbers appear more often in colloquial, non-scientific speech in the training text, hence biasing the LLMs. As for the bias towards 0.0 and 10.0, we speculate that this is because “0” and “1” are especially privileged numbers in the training data since they may also occur as boolean or binary symbols.
>
> As also observed by reviewer acu3, in the low-data limit, the estimated distribution is slightly skewed towards the lower end. This could be attributed to Benford's Law, which is the empirical observation that lower digits appear more frequently as leading digits in real-life data.
>
> It is quite striking, in our opinion, that LLMs could both inherit such biases induced by training data statistics, and also at the same time perform the unbiased KDE algorithm in the large data limit. Although such superposition likely makes a clean mechanistic interpretation very challenging.
>
> **Additional hints for adaptive kernel shape**
>
> We appreciate the insightful remark! In the low-data limit, one could indeed eyeball the kernel shape by studying the predicted density distribution. This was actually one of our motivations to parametrize the bespoke kernel in the way we did - so that at some value of s, the kernel shape could imitate the thick-tailed distribution as the reviewer observed in Figure 2 b.
>
> **Impact of $\alpha$ in Bayesian histogram**
>
> We thank the reviewer for the suggestion.
> Please note that the Bayesian histogram is not biased towards the general uniform distribution family, but a particular uniform distribution over the range 0 to 10. The target uniform distribution, however, is much narrower, roughly over the range 4 to 6, as shown in Figure 4. We think this discrepancy, along with the high alpha value (1) explains the slow convergence.
>
> According to our earlier experiments, setting alpha to 0 leads to erratic trajectories in the low data limit. This is because any new data point would update the histogram dramatically in the presence of prior. A strong uniform prior therefore not only makes Bayesian histogram start at similar points (ignorance) as the LLMs, but also makes the resulting trajectories smoother and easier to interpret, that is, following the Geodesics.
>
> In our updated appendix, we will visualize the histogram algorithm at different levels of alpha, in order to help the reader better understand the role played by uniform prior.
>
> **Missing citation**
>
> We thank the reviewer for bringing this to our attention. We will update our manuscript to correct this typo. The intended citation here is a textbook on Gaussian Processes:
> Matthias Seeger. Gaussian processes for machine learning. International journal of neural systems,
> 14(02):69–106, 2004.

---

> > ### Author Response · Authors · 2024-11-23
> > **Thank you, and looking forward to the reviewer's response**
> >
> > We thank the reviewer for the suggestions and valuable insights, which have helped up to substantially strengthen our paper. We hope that our added experiments, visualizations, and explanations have adequately addressed your questions and concerns.
> >
> > As the author-reviewer discussion period is coming to an end, we would be happy to read their reply to answer any additional questions. After that, we may not have the opportunity to respond to your comments.
> >
> > If you believe we have addressed your comments, we would be grateful if you could acknowledge this and update your score. Otherwise, we are happy to provide further clarification.
> >
> > Thank you again for your time and valuable review.

---

### Official Review · Reviewer_mikv · 2024-11-08

**Soundness:** 3
**Presentation:** 4
**Contribution:** 3
**Rating:** 8
**Confidence:** 3

**Summary:**

This paper is looking at the classical problem of density estimation with the use of large language models. The question of interest here is how the large language model estimates a density by relying on the data accessible to it "in context". A simple experiment is used to validate the proposal, namely, checking whether presenting an LLM (in this case, the LLaMA-2) with a sequence of points sampled iid from an underlying distribution results in the *next* sampled point eventually converge to come from the true distribution (with the increase of the number of observed data points).

This, of course, brings the question of how exactly does convergence of this process occur. The authors propose looking at low dimensional characteristics of the density estimate trajectories. Heuristically, at least, it appears that the LLM performs a kind of adaptive Gaussian estimation.

**Strengths:**

I think the question is a nice one and this is certainly (at least to me) an interesting paper on density estimation. It is not obvious a priori that the LLM will be able to do density estimation as it does. The trajectory analysis is certainly interesting. I would think this paper can allow us to do density estimation in automated ways. The experiments are well-designed for the most part.

**Weaknesses:**

I wonder what properties of the density are important here. I don't believe checking on a Uniform is sufficient - uniform densities are easy to estimate with a flexible kernel and indeed that is what the experiments show, this is not surprising. What is more of interest is to try a heavy-tailed example like a Cauchy or a t with a small number of degrees of freedom.

**Questions:**

What about even the bivariate normal case? The biggest issue with kernel density estimation methods is scaling with dimensionality. The refinement scheme seems sensible and easily adapted to the bivariate case, so would it be possible to run this scheme on bivariate normal data to see how well learning happens there. How are the correlations learned, for example?

---

> ### Author Response · Authors · 2024-11-17
>
> LLM’s in-context capabilities for PDF estimation. We provide below detailed poiny-by-point answers to the reviewer’s questions.
>
> **Heavy-tailed target distributions**
>
> We agree with the reviewer that heavy-tailed distributions are more challenging with classical Kernel methods. Following the reviewer’s suggestions, we are currently running additional experiments with Cauchy and t-distributions, and will share the results later during the rebuttal period.
>
> In Appendix A.7 of the current manuscript, we included experiments on randomly generated target PDFs. The in-context DE process of these are also well-described by the adaptive kernel method.
>
> **Generalization to multivariate density estimation**
>
> It would indeed be interesting to see how much the insights from this paper could generalize to multivariate time series, and indeed, we had thought about it too. However, using pre-trained LLMs to predict multivariate time-series systems is a non-trivial task, as explored in [1]. In particular, current state-of-the-art models struggle with order invariance. For example, the statistical prediction for the 2D tuple (A, B) might differ from that of (B, A). In other words, the formatting of multivariate data thus impacts the probabilistic prediction from LLMs.
>
> Generalizing the Hierarchy-PDF algorithm to the multivariate case could be useful for many future investigations. However, due to the difficulties explained above, we suspect such extension may not be as straightforward as at first sight. We therefore prefer to leave multivariate generalization to future work.
>
> [1]: Requeima, J., Bronskill, J.F., Choi, D., Turner, R.E., & Duvenaud, D.K. (2024). LLM Processes: Numerical Predictive Distributions Conditioned on Natural Language. ArXiv, abs/2405.12856.

---

> > ### Comment · Reviewer_mikv · 2024-11-26
> >
> > Thanks for adding in the additional experiments. It is good to see the method working in that context also.

---

### Author Response · Authors · 2024-11-17
**General response to all reviewers**

We would like to thank all reviewers for the detailed review and feedback on our work as well as the numerous suggestions. We are pleased to see that all four reviewers appreciated the novel approach of this work on exploring the ability of LLMs for probability density estimation, the significance of the problem, and that the paper is well-written and clear.

In the following we summarize the suggestions by reviewers, and the additional experiments we are currently running. Since these experiments are computationally intensive, we will be updating the manuscript with these throughout the discussion period.

Reviewers acu3, mikv, and 2CMq suggested additional experiments on heavy-tailed, multimodal, and skewed target distribution, such as Cauchy, Gaussian mixtures and t-distrbutions. They also suggested repeating some of the experiments using other LLMs, in order to expand the scope of our work. We will reproduce Figure 16 using said target distributions, and replot Figure 2. Part C using Mistral and Gemma - LLMs that are compatible with the hierarchy-PDF algorithm.

Reviewer KbiX suggested investigating the impact of number of digits (precision) on estimated densities. They also suggested studying the convergence when learning high-curvature random target PDFs, which can be achieved by extending the number of in-context data, and plot the Hellinger loss as a function of context length (n). We will show this loss curve in the updated Figure 34 of the appendix. Finally, they suggested visualizing the Bayesian histogram as different levels of alpha. We will show these visualization in a new section in the Appendix titled “Additional notes on Bayesian Histogram.”

Reviewer acu3 suggested additional visualizations using conventional loss functions, such as the (symmetrized) KL-divergence.

---

> ### Author Response · Authors · 2024-11-21
> **Update on additional experiments**
>
> The manuscript is now updated with new sections in the appendix that address some of the suggestions raised by reviewers:
>
> **A.4 In-context DE trajectories of LLaMA, Gemma, and Mistral**
> Documents additional experiments on three other recently released LLMs: Gemma-2b
> and -7b and Mistral-7b-v0.3. LLaMA, Gemma, and Mistral are three different suites of models built independently by different teams. Yet, they all demonstrate similar in-context DE trajectories, all of which could be well-approximated by the bespoke KDE method.
>
> **A.6 InPCA embeddings with symmetrized KL-divergence and Bhattacharyya distance**
> For completeness, we repeat InPCA embedding with other divergence measures.
> Unlike L2 embedding (vanilla PCA), the resulting trajectories appear smooth. However, they are difficult to interpret due to the presence of imaginary distances.
>
> We are now working on other additional experiments suggested by reviewers, and will update as the results come.

---

> ### Author Response · Authors · 2024-11-22
>
> The manuscript is updated with a new section in the appendix:
>
> **A.10 HEAVY-TAILED DISTRIBUTIONS**
> which visualizes the learning trajectories with heavy-tailed distributions. These experiments are performed with LLaMA as well as Mistral-v0.3 for generality.

---

> ### Author Response · Authors · 2024-11-23
> **Update on additional experiments and appendix**
>
> Dear all reviewers,
>
> The manuscript is updated with new sections in the appendix:
>
> **A.11 ADDITIONAL NOTES ON BAYESIAN HISTOGRAM**
>
> Visualizes the Bayesian histogram algorithm at different levels of prior bias ($\alpha$).
>
> **A.12 UNCERTAINTY QUANTIFICATION FOR BESPOKE KDE PARAMETERS**
>
> Explains in detail how the uncertainty in fitted KDE parameters are estimated based on information theory.

---

### Meta-Review · Area_Chair_XrzN · 2024-12-22

**Metareview:**

This is an interesting paper that studies the capability of LLMs to perform density estimation when prompted with iid samples from a unknown distribution interpreting next token probabilities as an estimated PDF that can be interpreted as a kind of adaptive Gaussian estimation.   The papers strengths are that it attempts to shed insight into intriguing properties of in-context learning for a classical statistical task. Weaknesses that were raised were: the distributions studied are possibly too simple e.g.1D problems; and the insights derived may be overfitting only one family/instance of models.  That said, there was unanimous agreement that the paper is well written, well motivated and provides some interpretability through clear experiments on how LLMs can solve a classical task given in-context examples.

**Additional Comments On Reviewer Discussion:**

The authors satisfactorily addressed most of the comments raised during the reviews. This is very well summarized in the last thread, so I wont repeat it here.

---

### Decision · Program_Chairs · 2025-01-22

Accept (Poster)